# Correcting Overparameterization Effects in Fair Empirical Risk Minimization

**Xiaoyi Mai** [1]    **Jean-Michel Loubes** [2]

## Abstract

Bias mitigation is particularly challenging for overparameterized machine learning (ML) models. Overfitting of training points not only amplifies data bias induced by spurious correlations, but also causes the failure of usual bias mitigation methods. To provide actionable insights to address this challenge, we propose a precise analysis of fair empirical risk minimization (ERM) in the overparameterized regime. Importantly, we show that even though conventional fair ERM fails on overparameterized models, this approach can be corrected by modifying the equality fairness constraint to allow for bias overcompensation. Moreover, our analysis presents an empirical criterion for strong equalized odds: balanced group-conditional means of representer coefficients, indicating equal average contribution from each sensitive group. Motivated by this result, we provide an estimable search interval that localizes the required overcompensation level for balanced coefficients. Despite the asymptotic nature of our findings, they closely capture the statistical behavior of moderately large ML models.

## 1. Introduction

With the widespread application of machine learning (ML) methods, a major concern arises about whether these data-driven automated decision processes exhibit fair behavior towards various population groups (Mehrabi et al., 2021; Besse et al., 2022). ML algorithms were found to not only perpetuate, but also exacerbate the bias already present in the training data set (Shah et al., 2020; Hooker, 2021; Wang & Russakovsky, 2021; Bell & Sagun, 2023; Bachoc et al., 2025). This causes the trained predictive model to make decisions that further penalize disadvantaged subpopulations.

[1]Univ Toulouse, UT2J, IMT, Toulouse, France [2]Regalia Team INRIA Bordeaux and ANITI Université de Toulouse. Correspondence to: Xiaoyi Mai <xiaoyi.mai@math.univ-toulouse.fr>, Jean-Michel Loubes <jean-michel.a.loubes@inria.fr>.

*Proceedings of the 43rd International Conference on Machine Learning*, Seoul, South Korea. PMLR 306, 2026. Copyright 2026 by the author(s).

The problem of algorithmic bias is more severe in overparameterized learning models, ubiquitous in modern ML with the success of deep neural networks (DNNs). On the one hand, overparameterization contributes to a better generalization performance under the double-descent phenomenon (Belkin et al., 2019; Bartlett et al., 2020; Nakkiran et al., 2021; Hastie et al., 2022; Mei & Montanari, 2022), where generalization performance improves as the model capacity increases beyond the interpolation threshold, at which the training error reaches zero. On the other hand, overparameterized models were found to amplify data bias by exacerbating spurious correlations in the training set (Hall et al., 2022; Sagawa et al., 2020; Subramonian et al., 2024; Bombari & Mondelli, 2025), causing worse fairness violation than their underparameterized counterparts.

Moreover, since overfitting causes overparameterized models to reproduce the labels of training data, it creates an illusion of fairness by producing the same zero training error for all groups. Consequently, bias mitigation techniques that train fair models through fairness-constrained empirical risk minimization (ERM) (Donini et al., 2018; Prost et al., 2019; Zafar et al., 2019; Oneto et al., 2020; Padala & Gujar, 2020; Perez-Suay et al., 2023) are rendered ineffective in the overparameterized regime, as empirically reported by (Cherepanova et al., 2021; Veldanda et al., 2023). In this work, we take a closer look at the difficulty of training fair large models by precisely characterizing the behavior of overparameterized fair ERM in a proportional large sample regime. Our analysis formalizes the issue of fair ERM when applied to large learning models and sheds critical insight on how to restore its effectiveness under overparameterization.

### 1.1. Fair ERM in the Overparametrized Regime

Consider data instance $(\mathbf{x}, \mathbf{y}, \mathbf{s})$ where $\mathbf{x} \in \mathbb{R}^d$ is a vector of predictors, $\mathbf{y} = \{0, 1\}$ the target label, and $\mathbf{s} = \{0, 1\}$ a binary sensitive attribute such as gender or race. Upon a biased training set of $n$ observations $\{(\mathbf{x}_i, \mathbf{y}_i, \mathbf{s}_i)\}_{i=1}^{n}$ with a positive correlation in the pairs $\{(\mathbf{y}_i, \mathbf{s}_i)\}_{i=1}^{n}$, a classifier $f_{\hat{\boldsymbol{\theta}}}$ is learned through ERM

$$\hat{\boldsymbol{\theta}} = \arg\min_{\boldsymbol{\theta}} \frac{1}{n} \sum_{i=1}^{n} \ell(f_{\boldsymbol{\theta}}(\mathbf{x}_i), \mathbf{y}_i), \tag{1}$$

for some convex loss $\ell \colon \mathbb{R} \times \{0,1\} \mapsto \mathbb{R}_{\geq 0}$. Replicating the positive correlation between $y_i$ and $s_i$, the resulting ERM classifier $f_{\hat{\boldsymbol{\theta}}}$ likely predicts higher score for the group $s = 1$.

Fair ERM adapts the training objective to integrate a desired fairness property (Verma & Rubin, 2018). Here we focus on strong equalized odds, defined as conditional independence of the classifier score $f_{\hat{\boldsymbol{\theta}}}(\mathbf{x})$ given the target y w.r.t the sensitive variable s, which requires

$$P(f_{\hat{\boldsymbol{\theta}}}(\mathbf{x}) > \tau | y, s) = P(f_{\hat{\boldsymbol{\theta}}}(\mathbf{x}) > \tau | y), \quad \forall \tau \in \mathbb{R}, \quad (2)$$

where $(\mathbf{x}, y, s)$ is independent of $\{(\mathbf{x}_i, y_i, s_i)\}_{i=1}^n$. As verifying Eq. (2) is computationally expensive for continuous score $f_{\hat{\boldsymbol{\theta}}}(\mathbf{x})$, fair ERM methods (Donini et al., 2018; Prost et al., 2019; Zafar et al., 2019; Oneto et al., 2020; Padala & Gujar, 2020) consider rather a weaker equality condition of form $\mathbb{E}[g(f_{\hat{\boldsymbol{\theta}}}(\mathbf{x}), y) | y, s] = \mathbb{E}[g(f_{\hat{\boldsymbol{\theta}}}(\mathbf{x}), y) | y]$ for some $g : \mathbb{R}^2 \mapsto \mathbb{R}$. The fair ERM problem is thus formulated as solving the original ERM in Eq. (1) under the constraint

$$\widehat{\mathbb{E}}[g(f_{\hat{\boldsymbol{\theta}}}(\mathbf{x}), y) | y, s] = \widehat{\mathbb{E}}[g(f_{\hat{\boldsymbol{\theta}}}(\mathbf{x}), y) | y], \quad (3)$$

where $\widehat{\mathbb{E}}$ denotes the sample mean on the training set, that is, for an arbitrary variable a, $\widehat{\mathbb{E}}[a | y \in Y, s \in S] := \frac{\sum_{i=1}^n a_i \mathbf{1}_{y_i \in Y} \mathbf{1}_{s_i \in S}}{\sum_{i=1}^n \mathbf{1}_{y_i \in Y} \mathbf{1}_{s_i \in S}}, \forall Y, S \subseteq \{0,1\}$.

When square loss $\ell(f_{\boldsymbol{\theta}}(\mathbf{x}), y) = (f_{\boldsymbol{\theta}}(\mathbf{x}) - y)^2$ is used, it is easy to see that fair ERM becomes ineffective in the overparametrized regime: as square loss produces a perfect match $f_{\hat{\boldsymbol{\theta}}}(\mathbf{x}_i) = y_i$ for all training point $(\mathbf{x}_i, y_i)$ above the interpretation threshold, the empirical fairness constraint in Eq. (3) is already met by the unconstrained ERM solution. This reasoning is in line with the empirical finding of Veldanda et al. (2023) that the ineffectiveness issue of fair ERM under overparameterization persists with logistic loss.

### 1.2. Our Contributions

Motivated by the ineffectiveness of fair ERM in the overparameterized regime, we propose to study a general framework of fair ERM that allows for an overcompensation or undercompensation of the training data bias, as presented in Section 3.1. We use linear loss $g(f_{\hat{\boldsymbol{\theta}}}(\mathbf{x}), y) = f_{\hat{\boldsymbol{\theta}}}(\mathbf{x}) - y$ for the fairness constrain in Eq. (3), a choice advocated by Donini et al. (2018); Oneto et al. (2020) to preserve the convexity of the ERM problem. Indeed, as discussed in Section 3.2, when $f_{\boldsymbol{\theta}}(\cdot)$ lives in a reproducing kernel Hilbert space equipped with kernel $k$, the fair ERM solution is given as $f_{\hat{\boldsymbol{\theta}}}(\cdot) = \frac{1}{n} \sum_{i=1}^n \hat{\alpha}_i k(\mathbf{x}_i, \cdot) + \hat{b}$ with the representer coefficients $\{\hat{\alpha}_i\}_{i=1}^n$ and the offset $\hat{b}$ determined by convex optimization under linear constraints. Note that the coefficient $\hat{\alpha}_i$ reflects the impact of the training point $\mathbf{x}_i$ on the prediction, through its proximity evaluated by $k$

Similarly to the settings of Sagawa et al. (2020); Wald et al. (2022), our data model, introduced in Section 3.3, assumes

the existence of spurious features correlated with the sensitive attribute s. To study the impact of overparameterization on fair ERM, we provide a sharp characterization of training and test scores, which is accessible through the representer coefficients $\{\hat{\alpha}_i\}_{i=1}^n$. From this main result presented in Section 4, several key conclusions emerge as laid out below.

- We discover an empirical criterion for achieving equalized odds that requires the average of the represent coefficients $\{\hat{\alpha}_i\}_{i=1}^n$ to be identical over the groups, i.e., $\widehat{\mathbb{E}}[\hat{\alpha} | s = 0] \simeq \widehat{\mathbb{E}}[\hat{\alpha} | s = 1]$. As discussed in Section 5.1 and stated in Corollary 5.1, having balanced coefficients, which can be understood as equal group contributions to the prediction, is a necessary and sufficient condition for equalized odds defined in Eq. (2).

- We find that the classical fair ERM approach, which enforces fairness on the training set, cannot generally train informative classifiers that satisfy equalized odds, as achieving balanced coefficients under this approach forces degeneration toward trivial predictions. Restoring the effectiveness of fair ERM in the overparameterized regime requires actually an overcompensation of the training data bias. These two results are developed in Section 5.2 and formalized in Theorem 5.2.

- Capitalizing on the above results, we offer in Algorithm 1 an implementation of the bias overcompensation method, and in Algorithm 2 a principled strategy to determine the required level of bias overcompensation with coverage guarantee given in Theorem 5.3. Guided by the empirical criterion of balanced coefficients, the proposed algorithm provably achieves equalized odd without using a holdout set for hyperparameter tuning.

From the discussion in Section 5, we know that balanced coefficients $\{\hat{\alpha}_i\}_{i=1}^n$, when achieved with bias overcompensation, guarantee equalized odds. In Section 6, we further investigate whether this equivalence between balanced coefficients and equalized odds holds more generally. Interestingly, our analysis reveals a negative answer to this question by showing that imposing directly balanced coefficients as an optimization constraint does not give rise to equalized odds. This finding calls for caution when it comes to modifying the optimization problem for an intended outcome, particularly in the overparameterized regime.

## 2. Related Work

**Fair ERM** Among various definitions of fairness proposed over the years, we find notably demographic parity and equalized odds (Verma & Rubin, 2018). The former refers to the same success rate predicted by $f_{\hat{\boldsymbol{\theta}}}$ for all groups and the latter the same prediction error. To meet the chosen

fairness requirement, one may resort to pre-processing, in-processing, or post-processing techniques, which involve, respectively, intervention on the training data, the training process, or the predicted outcome (Hort et al., 2024). Fair learning via in-processing can be carried out by altering the training objective (Donini et al., 2018; Agarwal et al., 2018; Hashimoto et al., 2018; Prost et al., 2019; Zafar et al., 2019; Sagawa et al., 2019; Oneto et al., 2020; Padala & Gujar, 2020; Do et al., 2022) or through adversarial training (Beutel et al., 2017; Zhang et al., 2018; Madras et al., 2018). In this branch, fair ERM refers to the family of methods that incorporate the fairness requirement as constraints or penalties. A key challenge of fair ERM is to define computationally tractable surrogates for measuring fairness, to replace the non-smooth and non-convex zero-one classification loss $g(f_{\hat{\theta}}(\mathbf{x}), y) = |\mathbf{1}_{>\tau}(f_{\hat{\theta}}(\mathbf{x})) - y|$ (Donini et al., 2018; Prost et al., 2019; Zafar et al., 2019; Oneto et al., 2020; Padala & Gujar, 2020; Do et al., 2022). For example, Donini et al. (2018) proposed a linear loss $g(f_{\hat{\theta}}(\mathbf{x}), y) = f_{\hat{\theta}}(\mathbf{x}) - y$, which forces a first-order match on training scores and results in a convex problem. This formulation is related to the Mindoff loss advanced by Prost et al. (2019), which evaluates the difference between the group score means in a kernel space. The first-order match is also connected to the idea of Beutel et al. (2019) to use correlation as a fairness metric. Most fair ERM methods are justified through population-level reasoning and do not ensure fairness in the overparameterized regime, where the empirical risk drastically differs from the expected risk.

**Bias amplification & mitigation under overparameterization** Recently, overparameterized models were found to amplify bias resulting from spurious correlations (Hall et al., 2022; Sagawa et al., 2020; Subramonian et al., 2024; Bombari & Mondelli, 2025). Intuitively, in the overparameterized regime, where the number of training instances is smaller than the dimension of learning model, it lacks sufficient data information to recover the target prediction function, causing the learning model to rely on spurious correlations. Hall et al. (2022) empirically confirmed the tendency of large models to rely more on the group membership s when it is easier to learn than the class membership y. Subsequently, Sagawa et al. (2020) theoretically analyzed this behavior in a Gaussian-mixture model with spurious correlations. The exacerbation of spurious correlations by overparameterized models was also demonstrated by the precise analyses of Subramonian et al. (2024); Bombari & Mondelli (2025) in the setting of linear regression. These analyses shed light on how to reduce the effect of bias amplification through model design choices (Subramonian et al., 2024; Bombari & Mondelli, 2025), or by downsampling majority groups (Sagawa et al., 2020). Meanwhile, when bias manifests itself in other forms than spurious correlations, overparameterization may help improve prediction

performance for minority groups (Roy et al., 2025).

The effect of fair learning techniques on overparameterized models is a less developed topic in comparison. Cherepanova et al. (2021) illustrated through experiments the tendency of large models to overfit to fairness constraints. Veldanda et al. (2023) showed that MinDiff (Prost et al., 2019), a principal fair ERM method, did not provide an effective fair learning of overparameterized models. These empirical observations align with the ineffectiveness of fairness constraints on interpolating solutions, as discussed in Section 1.1, and the inability of interpolating solutions to provide fair prediction according to the theoretical finding of Wald et al. (2022). Using a holdout set to calculate the fairness penalty might mitigate the overfitting problem to some extent (Zietlow et al., 2022; Dutt et al., 2024), but in this case the optimization solution may overfit the penalty on the holdout set (Cherepanova et al., 2021), resulting once again in a fairness generalization gap for new test data. When the in-processing approach fails, post-processing techniques are often used to enhance the fairness of overparameterized models (Menon et al., 2020; Alabdulmohsin & Lucic, 2021).

**Precise analyses of modern ML** Technically, the present work follows a long line of precise performance analyses in a proportional regime where the learning model dimension and the training sample size are comparably large (El Karoui et al., 2013; Donoho & Montanari, 2016; Huang, 2017; Thrampoulidis et al., 2018; Mai et al., 2019; Barbier et al., 2019; Mai et al., 2019; Gerace et al., 2020; Mai & Couillet, 2021; Taheri et al., 2021; Loureiro et al., 2022; Javanmard & Soltanolkotabi, 2022; Celentano et al., 2023).For trackablility, most precise analyses were performed under Gaussian or Gaussian mixture data models. Despite the restriction to specified data models, this approach is gaining importance for its adequacy to model modern ML. In particular, it offers characterizations of the double descent curve (Bartlett et al., 2020; Hastie et al., 2022; Mei & Montanari, 2022; Deng et al., 2022) that explain the generalizability of DNNs.

Our study shares a common interest in ML fairness with the recent analyses of Subramonian et al. (2024); Bombari & Mondelli (2025) on bias amplification. In contrast to the linear regression setting considered in these prior studies where the marginal distribution of $\mathbf{x}$ is not informative of y, our analysis addresses a classification setting similarly to Sagawa et al. (2020); Wald et al. (2022), where the values of y, s give rise to different patterns in $\mathbf{x}$. Our analysis also differs by providing a deep understanding of fair learning techniques in the overparaterized regime.

## 3. Problem Setup

We first introduce, in Section 3.1, our framework with tunable bias compensation that generalizes the standard fair

ERM method of Donini et al. (2018). Then in Section 3.2, we show that this generalized fair ERM problem can be solved by convex optimization through reformulation via representer theorems. Our data setting, which is specified in Section 3.3, considers independent features that are causally correlated with the target label y, or spuriously correlated with the protected variable s. Although these spurious correlations are effectively handled by the classical fair ERM method of Donini et al. (2018) in the heavily underparameterized regime, as we shall see in our analysis, their effect persists in the overparameterized regime.

### 3.1. Generalized Fair ERM with Tunable Bias Compensation

As in the methods of Donini et al. (2018); Oneto et al. (2020), we take $g(f_{\hat{\theta}}(\mathbf{x}), y) = f_{\hat{\theta}}(\mathbf{x}) - y$ in the empirical fairness constraint of Eq. (3) to obtain an equality constraint on average scores: $\Delta\widehat{\mathbb{E}}[f_{\theta}(\mathbf{x})|y = y, s] = 0$, $\forall y \in \{0, 1\}$, where $\Delta\widehat{\mathbb{E}}[f_{\theta}(\mathbf{x})|y = y, s] := \widehat{\mathbb{E}}[f_{\theta}(\mathbf{x})|y = y, s = 1] - \widehat{\mathbb{E}}[f_{\theta}(\mathbf{x})|y = y, s = 0]$. This fairness penalty is recovered as a special case of the MinDiff method (Prost et al., 2019) when MMD is evaluated by the inner product kernel. Aside from promising empirical results, this choice offers an important computational advantage: in the kernel space, Eq. (3) translates into linear constraints on the representer coefficients, as presented in Section 3.2.

As classical fair ERM become ineffective in the overparameterized regime, we propose a generalized fair ERM framework featuring a fairness constraint with tunable compensation: $\forall y \in \{0, 1\}$,

$$\Delta\widehat{\mathbb{E}}[f_{\theta}(\mathbf{x})|y = y, s] + \omega_y \Delta\widehat{\mathbb{E}}[f_{\theta}(\mathbf{x})|y] = 0, \quad (4)$$

where $\Delta\widehat{\mathbb{E}}[f_{\theta}(\mathbf{x})|y] := \widehat{\mathbb{E}}[f_{\theta}(\mathbf{x})|y = 1] - \widehat{\mathbb{E}}[f_{\theta}(\mathbf{x})|y = 0]$, and $\omega_0, \omega_1 \in \mathbb{R}$ are hyperparameters that adjust the bias compensation. With $\omega_0, \omega_1 = 0$, we retrieve the classical equality constraint with exact bias compensation. Note that $\Delta\widehat{\mathbb{E}}[f_{\theta}(\mathbf{x})|y]$ is the average score of the positive samples with $y_i = 1$ minus the average score of $y_i = 0$, which is normally greater than zero. Therefore, positive $\omega_0, \omega_1$ overcompensate the training data bias by imposing a higher empirical mean of the predicted scores for the underprivileged group $s_i = 0$, while negative $\omega_0, \omega_1$ enforce undercompensation.

### 3.2. Reformulation via Representer Theorems

The classification model is considered to take a general form $f_{\theta}(\cdot) = h_w(\cdot) + b$ with offset $b \in \mathbb{R}$ and mapping $h_w : \mathbb{R}^d \mapsto \mathbb{R}$ defined on a reproducing kernel Hilbert space $\mathcal{H}$ equipped with kernel $k : \mathbb{R}^d \times \mathbb{R}^d \to \mathbb{R}$ such that $\langle h_w(\cdot), k(\mathbf{x}, \cdot)\rangle_{\mathcal{H}} = h_w(\mathbf{x}), \forall h_w \in \mathcal{H}, \mathbf{x} \in \mathbb{R}^d$.

Let the model $f_{\theta}(\cdot)$ be trained by a ridge-regularized fair ERM that minimizes $\frac{1}{n}\sum_{i=1}^{n} \ell(f_{\theta}(\mathbf{x}_i), y_i) + \frac{\lambda}{2}\|h_w\|_{\mathcal{H}}^2$ for

some $\lambda > 0$, under the fairness constraint in Eq. (4). Then, by representer theorems, $f_{\hat{\theta}}$ is alternatively given as

$$f_{\hat{\theta}}(\cdot) = \frac{1}{n}\sum_{i=1}^{n}\hat{\alpha}_i k(\mathbf{x}_i, \cdot) + \hat{b} = \frac{1}{n}\hat{\boldsymbol{\alpha}}^{\mathsf{T}}k(\mathbf{X}, \cdot) + \hat{b} \quad (5)$$

with representer coefficient vector $\hat{\boldsymbol{\alpha}} := \{\hat{\alpha}_i\}_{i=1}^{n}$ and offset $\hat{b}$ determined by

$$\min_{(\boldsymbol{\alpha}, b) \in \mathbb{R}^n \times \mathbb{R}} \frac{1}{n}\sum_{i=1}^{n}\ell\left(\frac{1}{n}[\mathbf{K}\boldsymbol{\alpha}]_i + b, y_i\right) + \frac{\lambda}{2n^2}\boldsymbol{\alpha}^{\mathsf{T}}\mathbf{K}\boldsymbol{\alpha}$$

$$\text{s.t.} \quad \frac{\omega_y}{n}\left(\frac{\mathbf{q}_{10} + \mathbf{q}_{11}}{n_{10} + n_{11}} - \frac{\mathbf{q}_{00} + \mathbf{q}_{01}}{n_{00} + n_{01}}\right)^{\mathsf{T}}\mathbf{K}\boldsymbol{\alpha}$$

$$= \frac{1}{n}\left(\frac{\mathbf{q}_{y0}}{n_{y0}} - \frac{\mathbf{q}_{y1}}{n_{y1}}\right)^{\mathsf{T}}\mathbf{K}\boldsymbol{\alpha}, \quad \forall y \in \{0, 1\}, \quad (6)$$

where $\mathbf{K} := \{k(\mathbf{x}_i, \mathbf{x}_j)\}_{i,j=1}^{n}$, $\mathbf{q}_{ys} := \{\mathbf{1}_{y_i=y}\mathbf{1}_{s_i=s}\}_{i=1}^{n}$ and $n_{ys} := \|\mathbf{q}_{ys}\|_1$. To see the relation between the constraint in Eq. (6) and Eq. (4), notice that $f_{\theta}(\mathbf{x}_i) = \frac{1}{n}[\mathbf{K}\boldsymbol{\alpha}]_i + b$, and $\widehat{\mathbb{E}}[f_{\theta}(\mathbf{x})|y, s] = \frac{1}{nn_{ys}}\mathbf{q}_{ys}^{\mathsf{T}}\mathbf{K}\boldsymbol{\alpha} + b$, $\widehat{\mathbb{E}}[f_{\theta}(\mathbf{x})|y] = \frac{1}{n(n_{y0}+n_{y1})}(\mathbf{q}_{y0} + \mathbf{q}_{y1})^{\mathsf{T}}\mathbf{K}\boldsymbol{\alpha} + b$.

The fair ERM problem in Eq. (6) is a convex optimization problem under linear constraints. Let $\phi(\cdot) : \mathbb{R}^d \mapsto \mathbb{R}^p$ be the feature map such that $k(\mathbf{x}, \mathbf{x}') = \phi(\mathbf{x})^{\mathsf{T}}\phi(\mathbf{x}')$. In the overparameterized regime where the classification model dimension $p$ exceeds the training sample size $n$, Proposition 3.1 states the existence and uniqueness of solutions to Eq. (6).

**Proposition 3.1** (Uniqueness of representer coefficients)**.** *In the overparameterized regime of $p > n$, the optimization problem in Eq. (6) admits a unique solution $(\hat{\boldsymbol{\alpha}}, \hat{b})$ for linearly independent $\phi(\mathbf{x}_1), \ldots, \phi(\mathbf{x}_n) \in \mathbb{R}^p$.*

### 3.3. Data Model with Spurious Features

Recall that $\phi(\mathbf{x})$ is the feature vector of the data point $\mathbf{x}$ in the kernel space $\mathcal{H}$. Following the data settings of Sagawa et al. (2020); Wald et al. (2022), we consider the features in $\phi(\mathbf{x})$ to be causally correlated with y or spuriously correlated with s. Specifically,

$$\phi(\mathbf{x}) = \begin{bmatrix} y\boldsymbol{\mu}_{\mathcal{S}} \\ s\boldsymbol{\mu}_{\mathcal{S}^c} \end{bmatrix} + \mathbf{e} \in \mathbb{R}^p \quad (7)$$

where $\boldsymbol{\mu}_{\mathcal{S}}, \boldsymbol{\mu}_{\mathcal{S}^c}$ are deterministic vectors, and $\mathbf{e}$ is a centered random vector with independent entries (that are independent of y, s), of covariance $\boldsymbol{\Sigma} = \text{diag}(\{\sigma_l\}_{l=1}^{p})$. Since $\phi(\mathbf{x})_{\mathcal{S}^c}$ is independent of y when conditioned on s, $\mathcal{S}^c$ (resp. $\mathcal{S}$) is the index set of spurious (resp. causal) features.

The training data is biased in the sense that the individuals in the group $s_i = 1$ have a higher rate of positive labels $y_i = 1$ than those with $s_i = 0$. Mathematically speaking,

$$\gamma := \widehat{P}(y = 1|s = 1) - \widehat{P}(y = 1|s = 0) > 0. \quad (8)$$

where $\widehat{P}$ denotes the empirical probability estimated over the training set, e.g., $\widehat{P}(y|s) := \frac{n_{ys}}{n_{0s}+n_{1s}}$. A consequence of Eq. (8) is a positive correlation between $\{y_i\}_{i=1}^n$ and $\{s_i\}_{i=1}^n$, which would be learned through ERM, and result in a favorable prediction towards the group $s = 1$.

**Proposition 3.2.** *[Fairness in the heavily underparameterized regime] Let $f_{\hat{\theta}}(\cdot) = \frac{1}{n}\hat{\alpha}^\mathsf{T} k(\mathbf{X}, \cdot) + \hat{b}$ with $\hat{\alpha}, \hat{b}$ given by Eq. (6). Then, under Eq. (7), the equalized odds property in Eq. (2) is met if and only if $\Delta\mathbb{E}[f_{\hat{\theta}}(\mathbf{x})|y = y, s] = 0$.*

Proposition 3.2 implies that in the abundant sample regime where $n \gg p$ so that $\widehat{\mathbb{E}}[f_{\hat{\theta}}(\mathbf{x})|y, s] \to \mathbb{E}[f_{\hat{\theta}}(\mathbf{x})|y, s]$, equalized odds are achieved by imposing Eq. (4) with $\omega_0, \omega_1 = 0$.

# 4. Sharp Characterization of Fair ERM

Our analysis is conducted under the following assumptions.

**Assumption 4.1** (Training data). For the training set, the feature vectors $\{\phi(\mathbf{x}_i)\}_{i=1}^n$ are as described in Eq. (7) with i.i.d. $\mathbf{e}_{i\in[n]}$ of bounded fourth moment. Also, a data bias as formulated in Eq. (8) is present in the i.i.d. pairs of target and sensitive attributes $\{(y_i, s_i)\}_{i=1}^n$.

**Assumption 4.2** (Large overparameterized regime). As $n \to \infty$ with fixed $p/n, n_{ys}/n, |\mathcal{S}|/p \in (0, 1)$, for the data model in Eq. (7), we have that (i) $\|\boldsymbol{\mu}_\mathcal{S}\|_2, \|\boldsymbol{\mu}_{\mathcal{S}^c}\| = \Theta(1), \|\boldsymbol{\mu}_\mathcal{S}\|_\infty, \|\boldsymbol{\mu}_{\mathcal{S}^c}\|_\infty = \Theta(1/\sqrt{p})$, (ii) $\sigma_{l\in[p]} = \Theta(1)$.

Assumption 4.2 places us in an overparameterization regime with comparable $n > p$, commensurately large training subsets defined by the value of $(y, s)$, as well as comparable numbers of casual and spurious features. Item (i) indicates that there is sufficient, but not abundant, information in $\mathbf{x}$ to predict $(y, s)$, and this information is evenly distributed over numerous features. Item (ii) ensures that the feature vector $\phi(\mathbf{x})$ does not live in a smaller dimensional subspace of $\mathbb{R}^p$.

**Assumption 4.3** (Loss function). The loss function $\ell(r, y): \mathbb{R} \times \{0, 1\} \mapsto \mathbb{R}_{\geq 0}$ is strictly convex and three-times differentiable with respect to $r$. The minimum value of $\ell$ is only attained when $r = y$.

Assumption 4.3 is met for two predominant loss functions: logistic loss $\ell_{\mathrm{LR}}(r, y) = \log\left(1 + e^{(1-2y)r}\right)$ and square loss $\ell_{\mathrm{LS}}(r, y) = (r - y)^2$.

Our main theorem characterizes the distribution of the predicted score $f_{\hat{\theta}}(\mathbf{x})$ for a new data point $(\mathbf{x}, y, s)$, as well as the empirical distribution of the training scores $\{f_{\hat{\theta}}(\mathbf{x}_i)\}_{i=1}^n$, through the representer coefficients $\{\hat{\alpha}_i\}_{i=1}^n$. Let us first define the differences between the conditional empirical means of $\hat{\alpha}_i$:

$$\Delta\widehat{\mathbb{E}}[\hat{\alpha}|y] := \widehat{\mathbb{E}}[\hat{\alpha}|y = 1] - \widehat{\mathbb{E}}[\hat{\alpha}|y = 0],$$
$$\Delta\widehat{\mathbb{E}}[\hat{\alpha}|s] := \widehat{\mathbb{E}}[\hat{\alpha}|s = 1] - \widehat{\mathbb{E}}[\hat{\alpha}|s = 0],$$

which, as we shall see in Theorem 4.4, are crucially related to the accuracy and fairness of the classifier $f_{\hat{\theta}}$.

**Theorem 4.4** (Characterization of test and training scores). *Let Assumptions 4.1, 4.2 and 4.3 hold. Then, for $f_{\hat{\theta}}(\cdot) = \frac{1}{n}\hat{\alpha}^\mathsf{T} k(\mathbf{X}, \cdot) + \hat{b}$ with $\hat{\alpha}, \hat{b}$ given by Eq. (6) on a training set $\mathcal{D}_{\mathrm{train}} = \{\mathbf{x}_i, y_i, s_i\}_{i=1}^n$, there exists finite $\nu > 0$ such that, for any bounded Lipschitz $h: \mathbb{R} \mapsto \mathbb{R}$, we have*

$$\mathbb{E}[h(f_{\hat{\theta}}(\mathbf{x}))|y, s] - \mathbb{E}[h(\eta_{ys} + \epsilon)|y, s, \mathcal{D}_{\mathrm{train}}] \xrightarrow{P} 0, \quad (9)$$
$$\widehat{\mathbb{E}}[h(f_{\hat{\theta}}(\mathbf{x}) - \kappa\hat{\alpha})|y, s] - \mathbb{E}[h(f_{\hat{\theta}}(\mathbf{x}))|y, s] \xrightarrow{P} 0, \quad (10)$$

*where $\kappa = \frac{1}{n}\mathrm{Tr}\,\mathbf{Q}(\nu)\boldsymbol{\Sigma}$ with $\mathbf{Q}(\nu) := (\boldsymbol{I}_p + \nu\boldsymbol{\Sigma})^{-1}$,*

$$\eta_{ys} = y\boldsymbol{\mu}_\mathcal{S}^\mathsf{T}\tilde{\boldsymbol{\beta}}_\mathcal{S}\Delta\widehat{\mathbb{E}}[\hat{\alpha}|y] + s\boldsymbol{\mu}_{\mathcal{S}^c}^\mathsf{T}\tilde{\boldsymbol{\beta}}_{\mathcal{S}^c}\Delta\widehat{\mathbb{E}}[\hat{\alpha}|s] + \hat{b} \quad (11)$$

*with $\tilde{\boldsymbol{\beta}}^\mathsf{T} = \mathbf{Q}(\nu)\left[\prod_{y=0}^1 \widehat{P}(y = y)\boldsymbol{\mu}_\mathcal{S}^\mathsf{T} \prod_{s=0}^1 \widehat{P}(s = s)\boldsymbol{\mu}_{\mathcal{S}^c}^\mathsf{T}\right]$, and $\epsilon \sim \mathcal{N}(0, \frac{1}{n^2}\|\hat{\alpha}\|^2\|\mathbf{Q}(\nu)\boldsymbol{\Sigma}\|_2^2 + \tilde{\boldsymbol{\beta}}_\mathcal{S}^\mathsf{T}\boldsymbol{\Sigma}_{\mathcal{S}\mathcal{S}}\tilde{\boldsymbol{\beta}}_\mathcal{S}\Delta\widehat{\mathbb{E}}[\hat{\alpha}|y]^2 + \tilde{\boldsymbol{\beta}}_{\mathcal{S}^c}^\mathsf{T}\boldsymbol{\Sigma}_{\mathcal{S}^c\mathcal{S}^c}\tilde{\boldsymbol{\beta}}_{\mathcal{S}^c}\Delta\widehat{\mathbb{E}}[\hat{\alpha}|s]^2)$.*

Essentially, Theorem 4.4 tells us that in the limit of comparably large $n, p$, the test score $f_{\hat{\theta}}(\mathbf{x})$ has the same asymptotic distribution as $\eta_{ys} + \epsilon$ in the sense of Eq. (9), and the distribution of $\eta_{ys} + \epsilon$ depends on the statistics of the representer coefficients $\{\hat{\alpha}_i\}_{i=1}^n$ as detailed in Eq. (11). Theorem 4.4 also specifies a relation between the different distributions of the training and test scores, by showing in Eq. (10) that the empirical distribution of $\{f_{\hat{\theta}}(\mathbf{x}_i) - \kappa\hat{\alpha}_i\}_{i=1}^n$, which are the training scores minus a scaling of the represent coefficients, is asymptotically close to the distribution of the test score $f_{\hat{\theta}}(\mathbf{x})$.

Figure 1 offers a numerical validation of Theorem 4.4, where a close match is observed between theoretical prediction and practical reality for a moderately large data setting with $n = 5000$, $p = 3n$. As we can see from the orange histogram, the training scores $\{f_{\hat{\theta}}(\mathbf{x}_i)\}_{i=1}^n$ are perfectly separated by the class label $y$, due to the overfitting of the training points. In contrast, the histogram of $\{f_{\hat{\theta}}(\mathbf{x}_i) - \kappa\hat{\alpha}_i\}_{i=1}^n$ in blue nearly overlaps with the distribution of the test score $f_{\hat{\theta}}(\mathbf{x})$, both well predicted by the probability density of $\eta_{ys} + \epsilon$.

# 5. Consequences

From the sharp characterization in Theorem 4.4, we can deduce when equalized odds are achieved from the conditional averages of the representer coefficients $\{\hat{\alpha}_i\}_{i=1}^n$. We establish in Section 5.1 an equivalence between equalized odds and balanced group averages of representer coefficients. The discussion in Section 5.2 relates the failure of classical fair ERM to the unbalanced coefficients produced under this strategy, and points out the necessity of bias overcompensation. These results motivate a practical procedure

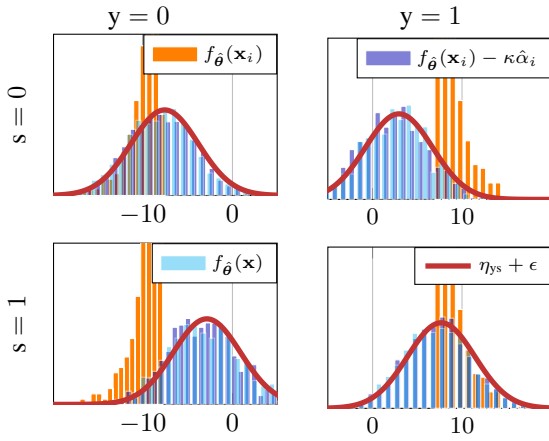

*Figure 1.* Distributions of scores versus characterization by Theorem 4.4, for $f_{\hat{\boldsymbol{\theta}}}$ given by Eq. (6) with $\ell = \ell_{\mathrm{LR}}$, $\lambda = 10^{-4}$, $\omega_0 = \omega_1 = 0$ on $n = 5000$ training instances drawn from Eq. (7) with $P(\mathrm{s} = 1) = 0.5$, $P(\mathrm{y} = 1|\mathrm{s} = 0) = 0.2$, $P(\mathrm{y} = 1|\mathrm{s} = 1) = 0.8$, $p = 3n$, $\frac{1}{4}\boldsymbol{\mu}_{\mathcal{S}}$, $\frac{1}{16}\boldsymbol{\mu}_{\mathcal{S}^{\mathrm{c}}} \sim \mathrm{Unif}(\mathbb{S}^{p/2-1})$, $\sigma_{l \in [p]} \sim \chi_1$, and uniform noises in $\mathbf{e}$. The histograms in orange and blue represent respectively the empirical distributions of $\{f_{\hat{\boldsymbol{\theta}}}(\mathbf{x}_i)\}_{i=1}^n$ and $\{f_{\hat{\boldsymbol{\theta}}}(\mathbf{x}_i) - \kappa\hat{\alpha}_i\}_{i=1}^n$ for $\kappa$ given by Theorem 4.4 with $\theta = 0.361$. The histogram of test scores in cyan is obtained from $n$ test examples. The red line plots the probability density of $\eta_{\mathrm{ys}} + \epsilon$ calculated from Theorem 4.4 with $\theta = 0.361$.

to achieve equalized odds through a bias overcompensation that induces balanced coefficients. Implemented in Algorithm 1 with coverage guarantee given in Theorem 5.3, this method effectively corrects the classical fair ERM approach, as demonstrated by the experiments in Section 5.3.

### 5.1. Fairness Attained at Balanced Coefficients

From Eq. (9), we observe that equalized odds as defined in Eq. (2) are achieved when $\eta_{\mathrm{y}0} \simeq \eta_{\mathrm{y}1}$, which occurs if and only if $\Delta\widehat{\mathbb{E}}[\hat{\alpha}|\mathrm{s}] \simeq 0$ according to Eq. (11). Thus, we have the equivalence between equalized odds and balanced coefficients stated in Corollary 5.1.

**Corollary 5.1** (Condition on representer coefficients for equalized odds). *Under the conditions and notations of Theorem 4.4, the property of equalized odds defined in Eq. (2) is satisfied asymptotically if and only if $\Delta\widehat{\mathbb{E}}[\hat{\alpha}|\mathrm{s}] \xrightarrow{P} 0$.*

### 5.2. Correcting Fair ERM with Bias Overcompensation

To achieve equalized odds, we require balanced coefficients with $\Delta\widehat{\mathbb{E}}[\hat{\alpha}|\mathrm{s}] \simeq 0$ according to Corollary 5.1. Note from Theorem 4.4 that under balanced coefficients, we have $\Delta\widehat{\mathbb{E}}[f_{\boldsymbol{\theta}}(\mathbf{x})|\mathrm{y} = y, \mathrm{s}] \simeq \kappa\Delta\widehat{\mathbb{E}}[\hat{\alpha}|\mathrm{y} = y, \mathrm{s}]$ where $\Delta\widehat{\mathbb{E}}[\hat{\alpha}|\mathrm{y} = y, \mathrm{s}] := \widehat{\mathbb{E}}[\hat{\alpha}|\mathrm{y} = y, \mathrm{s} = 1] - \widehat{\mathbb{E}}[\hat{\alpha}|\mathrm{y} = y, \mathrm{s} = 0]$. Then, the fairness constraint in Eq. (4) enforces that, $\forall y \in \{0, 1\}$,

$$\kappa\Delta\widehat{\mathbb{E}}[\hat{\alpha}|\mathrm{y} = y, \mathrm{s}] + \omega_{\mathrm{y}}\Delta\widehat{\mathbb{E}}[f_{\boldsymbol{\theta}}(\mathbf{x})|\mathrm{y}] \simeq 0. \quad (12)$$

As the classical fair ERM method applies exact bias compensation with $\omega_0, \omega_1 = 0$, it imposes $\Delta\widehat{\mathbb{E}}[\hat{\alpha}|\mathrm{y} = y, \mathrm{s}] \simeq 0$ for $y \in \{0, 1\}$ as a result of Eq. (12). Using $\widehat{\mathbb{E}}[\hat{\alpha}|\mathrm{s}] = \sum_{y=0}^1 \widehat{P}(\mathrm{y} = y|\mathrm{s})\widehat{\mathbb{E}}[\hat{\alpha}|\mathrm{y} = y, \mathrm{s}]$ and $\widehat{\mathbb{E}}[\hat{\alpha}|\mathrm{y}] = \sum_{s=0}^1 \widehat{P}(\mathrm{s} = s|\mathrm{y})\widehat{\mathbb{E}}[\hat{\alpha}|\mathrm{y}, \mathrm{s} = s]$, we obtain that

$$\Delta\widehat{\mathbb{E}}[\hat{\alpha}|\mathrm{s}] - \gamma\Delta\widehat{\mathbb{E}}[\hat{\alpha}|\mathrm{y}] = \sum_{y=0}^1 \zeta_y\Delta\widehat{\mathbb{E}}[\hat{\alpha}|\mathrm{y} = y, \mathrm{s}], \quad (13)$$

where $\zeta_{\mathrm{y}} := \frac{\prod_{s=0}^1 \widehat{P}(\mathrm{y}, \mathrm{s}=s)}{\widehat{P}(\mathrm{y})} \sum_{s=0}^1 \frac{1}{\widehat{P}(\mathrm{s}=s)}$. Since $\zeta_0, \zeta_1 > 0$, and $\gamma > 0$ as stated in Eq. (8), we notice from Eq. (13) that having $\Delta\widehat{\mathbb{E}}[\hat{\alpha}|\mathrm{s}] \simeq 0$ and $\Delta\widehat{\mathbb{E}}[\hat{\alpha}|\mathrm{y} = y, \mathrm{s}] \simeq 0$ for $y \in \{0, 1\}$ is not possible except when $\Delta\widehat{\mathbb{E}}[\hat{\alpha}|\mathrm{y}] \simeq 0$, in which case the classifier $f_{\hat{\boldsymbol{\theta}}}$ performs no better than random guess according to Eq. (9) and Eq. (11). In other words, the classical fair ERM method with exact bias compensation is bound to fail in the overparameterized regime, as it is unable to produce meaningful and fair classification.

Since canceling the training data bias with $\omega_{\mathrm{y}} = 0$ does not lead to fair results, we now study whether equalized odds can be achieved with overcompensation (i.e., $\omega_{\mathrm{y}} > 0$) or undercompensation (i.e., $\omega_{\mathrm{y}} < 0$). Plugging Eq. (12) into Eq. (13) with $\Delta\widehat{\mathbb{E}}[\hat{\alpha}|\mathrm{s}] \simeq 0$, we obtain

$$\zeta_0\omega_0 + \zeta_1\omega_1 - \frac{\kappa\gamma\Delta\widehat{\mathbb{E}}[\hat{\alpha}|\mathrm{y}]}{\Delta\widehat{\mathbb{E}}[f_{\boldsymbol{\theta}}(\mathbf{x})|\mathrm{y}]} \simeq 0,$$

which entails $\zeta_0\omega_0 + \zeta_1\omega_1 > 0$. Since $\zeta_0, \zeta_1 > 0$, at least one of $\omega_0, \omega_1$ is strictly positive. We conclude that bias overcompensation is required to fairly learn overparameterized models.

**Theorem 5.2** (Fairness in the overparameterized regime). *Under the same assumptions and notations as in Theorem 4.4, we have the following results.*

- Failure of fair ERM*: when setting $\omega_0, \omega_1 = 0$, if*

$$\mathbb{E}\big[h(f_{\hat{\boldsymbol{\theta}}}(\mathbf{x}))\big|\mathrm{y}, \mathrm{s}\big] - \mathbb{E}\big[h(f_{\hat{\boldsymbol{\theta}}}(\mathbf{x}))\big|\mathrm{y}\big] \to 0,$$

*we always have a trivial classification with*

$$\mathbb{E}\big[h(f_{\hat{\boldsymbol{\theta}}}(\mathbf{x}))\big|\mathrm{y}\big] - \mathbb{E}\big[h(f_{\hat{\boldsymbol{\theta}}}(\mathbf{x}))\big] \to 0.$$

- Necessity of bias overcompensation*: A fair and informative classification, in the sense that*

$$\mathbb{E}\big[h(f_{\hat{\boldsymbol{\theta}}}(\mathbf{x}))\big|\mathrm{y}, \mathrm{s}\big] - \mathbb{E}\big[h(f_{\hat{\boldsymbol{\theta}}}(\mathbf{x}))\big|\mathrm{y}\big] \to 0,$$
$$\liminf_{n \to \infty} \mathrm{Cor}(f_{\hat{\boldsymbol{\theta}}}(\mathbf{x}), \mathrm{y}) > 0, \quad (14)$$

*can only be achieved when at least one of $\omega_0, \omega_1$ is strictly positive.*

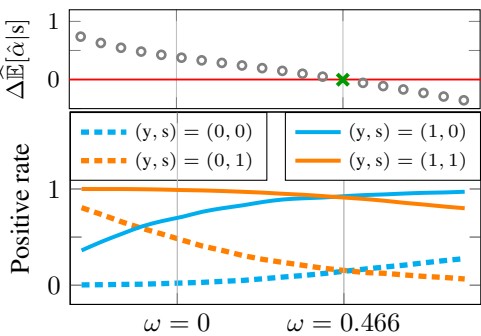

*Figure 2.* Gap between group average coefficients $\Delta\widehat{\mathbb{E}}[\hat{\alpha}|s]$ *(Top)*, and conditional positive rates $P(\hat{y} = 1|y, s)$ *(Bottom)*, as a function of $\omega_0 = \omega_1 = \omega$. Classification by logistic regression $\hat{y} = \mathbf{1}_{>0}(f_{\hat{\theta}}(\mathbf{x}))$ with $f_{\hat{\theta}}$ given by Eq. (6) for $\ell = \ell_{\mathrm{LR}}$ and $\lambda = 10^{-4}$, on $n = 500$ training instances drawn from Eq. (7) with $P(s = 1) = 0.5$, $P(y = 1|s = 0) = 0.2$, $P(y = 1|s = 1) = 0.8$, $p = 4n$, $\frac{1}{2}\boldsymbol{\mu}_{\mathcal{S}}, \frac{1}{8}\boldsymbol{\mu}_{\mathcal{S}^c} \sim \mathrm{Unif}(\mathbb{S}^{p/2-1})$, $12\sigma_{l\in[p]} \sim \mathrm{Unif}(\{1, 3, 5\})$, and Rademacher noises in $\mathbf{e}$. Performance evaluated on $10^4$ test examples.

Figure 2 provides a finite-dimensional illustration of Theorem 5.2 at $n = 500$ and $p = 4n$. In the upper part of Figure 2, we can see that $\Delta\widehat{\mathbb{E}}[\hat{\alpha}|s]$ diminishes as the compensation hyperparameters $\omega_0 = \omega_1 = \omega$ increase. The group true positive rates $P(\hat{y} = 1|y = 1, s = 0), P(\hat{y} = 1|y = 1, s = 1)$ are represented by smooth lines in the bottom of Figure 2, and the group false positive rates $P(\hat{y} = 1|y = 0, s = 0), P(\hat{y} = 1|y = 0, s = 1)$ by densely dashed lines. The failure of classical fair ERM is demonstrated by the significant gap between the group true positive rates at $\omega = 0$ and also the one between the group false positive rates, both of which exhibit a favorable bias towards the group $s = 1$. Moving to the zone of bias overcompensation with $\omega > 0$ reduces the gap in the true (or false) positive rates, which is closed near where $\Delta\widehat{\mathbb{E}}[\hat{\alpha}|s]$ reaches zero at $\omega = 0.466$. This observation confirms the equivalence between equalized odds and balanced coefficients predicted by Corollary 5.1.

### 5.3. Tuning the Level of Bias Overcompensation

The equivalence between equalized odds and balanced coefficients established in Corollary 5.1 allows for a principled tuning of bias compensation without cross validation, by searching for the values of $\omega_0, \omega_1$ that cancel $\Delta\widehat{\mathbb{E}}[\hat{\alpha}|s]$. Using sharp asymptotic results, we specify in Algorithm 2 an estimable search range for $\omega_0, \omega_1$, with coverage guarantee given in Theorem 5.3.

**Theorem 5.3** (Tuning bias compensation with coverage guarantee)**.** *Consider the same conditions and notations as in Theorem 4.4. Let $\rho_0, \rho_1, \omega_{\mathrm{inf}}, \omega_{\mathrm{sup}}$ be computed by Algorithm 2. If there exists $\omega \in \mathbb{R}$ such that the classifier $f_{\hat{\theta}}$ given by Eq. (6) with $\omega_y = \rho_y\omega$ for $y \in \{0, 1\}$ achieves a*

*non-trivial fair classification in the sense of Eq. (14), then we have $\omega \in [\omega_{\mathrm{inf}}, \omega_{\mathrm{sup}}]$ with high probability.*

Using $\rho_0, \rho_1, \omega_{\mathrm{inf}}, \omega_{\mathrm{sup}}$ returned by Algorithm 2, Algorithm 1 implements the bias overcompensation method by searching the value of $\omega$ on a grid defined by $\omega_{\mathrm{inf}}, \omega_{\mathrm{sup}}$ that minimizes $|\Delta\widehat{\mathbb{E}}[\hat{\alpha}|s]|$ with $\omega_y = \rho_y\omega$ for $y \in \{0, 1\}$. The hyperparameter $\upsilon \in \mathbb{R}_{\geq 0}$ expands the search range to account for finite-dimensional effects and violations of data assumptions, and $T \in \mathbb{Z}_{>0}$ controls the grid size.

As shown in the right column of Figure 3, the bias overcompensation method implemented by Algorithm 1 works effectively in the overparameterized regime, leading to a more accurate prediction without sacrificing fairness. Meanwhile, the bias amplification phenomenon persists under classical fair ERM, as illustrated in the left column of Figure 3, where the gap between the group true (or false) positive rates increases with the ratio $p/n$. These trends are replicated by the empirical results on Color-MNIST data (Arjovsky et al., 2019; Subramonian et al., 2024; Bombari & Mondelli, 2025) reported in Figure 4. We use the procedure of Bombari & Mondelli (2025) to create a binary problem for even ($y = 0$) and odd ($y = 1$) numbers on MNIST images colored blue ($s = 0$) or red ($s = 0$), and perturbed by a white Gaussian noise of variance 0.05. The parity y and the color s are spuriously correlated in the training set with $P(s = y) = 0.9$. Similarly to Mei & Montanari (2022), we use feature representations obtained by transformed random projections $\mathrm{ReLU}(\mathbf{W}^\top\mathbf{x})$ where $\mathbf{W} \in \mathbb{R}^{p\times d}$ has i.i.d. columns drawn from $\mathrm{Unif}(\mathbb{S}^{d-1})$.

## 6. Further Discussion: Is Fairness Achieved with Imposed Balanced Coefficients?

From Section 5, we know that balanced coefficients, when achieved with a bias overcompensation, always result in equalized odds. This raises a natural question as to whether equalized odds can be realized by an ERM subject to the constraint of balanced coefficients $\Delta\widehat{\mathbb{E}}[\hat{\alpha}|s] = 0$:

$$\min_{(\boldsymbol{\alpha}, b)\in\mathbb{R}^n\times\mathbb{R}} \frac{1}{n}\sum_{i=1}^{n}\ell\left(\frac{1}{n}[\mathbf{K}\boldsymbol{\alpha}]_i + b, y_i\right) + \frac{\lambda}{2n^2}\boldsymbol{\alpha}^\top\mathbf{K}\boldsymbol{\alpha}$$

$$\mathrm{s.t.}\quad \frac{1}{n}\left(\frac{\mathbf{q}_{00} + \mathbf{q}_{10}}{n_{00} + n_{10}} - \frac{\mathbf{q}_{01} + \mathbf{q}_{11}}{n_{01} + n_{11}}\right)^\top\boldsymbol{\alpha} = 0. \quad (15)$$

Similarly to Eq. (6), the above optimization is also a convex problem under linear constraints. As it forces balanced coefficients as an optimization constraint, there is no need for hyperparameter tuning. However, our result in Theorem 6.1 indicates that this approach does not ensure equalized odds.

**Theorem 6.1** (Ineffectiveness of enforcing balanced coefficients)**.** *Let Assumptions 4.1, 4.2 and 4.3 hold. Then, for $f_{\hat{\theta}}(\cdot) = \frac{1}{n}\hat{\boldsymbol{\alpha}}^\top k(\mathbf{X}, \cdot) + \hat{b}$ with $\hat{\boldsymbol{\alpha}}, \hat{b}$ given by Eq. (15), we*

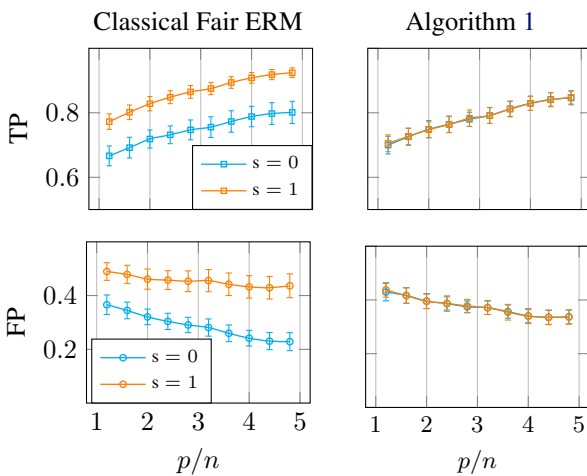

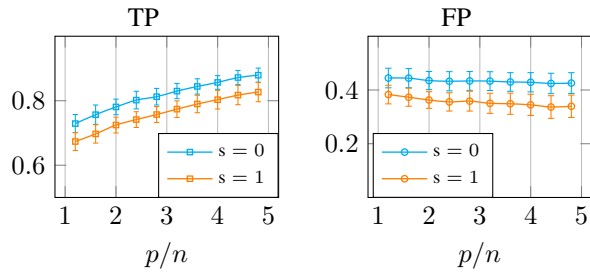

*Figure 5.* Group true positive rates $P(\hat{y} = 1|y = 1, s)$ *(Left)*, and group false positive rates $P(\hat{y} = 1|y = 0, s)$ *(Right)*, with $f_{\hat{\theta}}$ given by Eq. (15) under the same setting as in Figure 3.

*have*

$$\mathbb{E}\big[f_{\hat{\theta}}(\mathbf{x})\big|y, s\big] - \mathbb{E}\big[f_{\hat{\theta}}(\mathbf{x})\big|y\big] = \Theta(1). \qquad (16)$$

The negative result in Theorem 6.1 is confirmed by the experiments in Figure 5, conducted in the same setting as in Figure 3, but with the classifier $f_{\hat{\theta}}$ given by Eq. (15). In contrast to the fair results produced by the overcompensation method in the right column of Figure 3, the imposed balanced coefficients induce a reverse effect that penalizes the previously advantaged group $s = 1$, as shown in Figure 5.

# 7. Concluding Remarks

In this work, we provided a sharp, high-dimensional characterization of fairness-constrained empirical risk minimization in an overparameterized kernel regime. Our study was placed in a basic classification setting with spurious correlations, which mirrors many trends of bias amplification in practical ML according to Sagawa et al. (2020); Wald et al. (2022). Our analysis gave access to the asymptotic distributions of test and training scores via representer coefficients, which indicate contributions of training points. It showed that the conditional distribution of test score given the target label and the protected attribute has a Gaussian limit with group-independent variance, so the equality of distributions reduces to the equality of conditional means.

*Figure 3.* Group true positive rates $P(\hat{y} = 1|y = 1, s)$ *(Top)*, and group false positive rates $P(\hat{y} = 1|y = 0, s)$ *(Bottom)*, given by classical fair ERM with $\omega_0 = \omega_1 = 0$ *(Left)*, and the overcompensation method implemented in Algorithm 1 with $\upsilon = 0.2$, $T = 20$ *(Right)*. Classification by logistic regression $\hat{y} = \mathbf{1}_{>0}(f_{\hat{\theta}}(\mathbf{x}))$ with $f_{\hat{\theta}}$ given by Eq. (6) for $\ell = \ell_{LR}$ and $\lambda = 10^{-4}$, on $n = 500$ training instances drawn from Eq. (7) with $P(s = 1) = 0.6$, $P(y = 1|s = 0) = 0.3$, $P(y = 1|s = 1) = 0.8$, $\frac{1}{8}\boldsymbol{\mu}_{\mathcal{S}}, \frac{1}{8}\boldsymbol{\mu}_{\mathcal{S}^c} \sim \mathrm{Unif}(\mathbb{S}^{p/2-1})$, $\sigma_{l \in [p]} \sim \mathrm{Unif}(\{1, 5\})$, and uniform noises in $\mathbf{e}$. Performance evaluated on $10^4$ test examples, and averaged over 50 trials with $\pm 1$ standard deviation.

Our analysis has several implications. First, it traced the failure of classical fair ERM to its inability to produce groupwise balanced coefficients. Then the analysis suggested overcompensating the training data bias to induce balanced coefficients, which in turn give rise to equalized odds. Finally, we derived an estimable interval that localizes the required overcompensation level with provable coverage. Furthermore, we highlighted the interest of bias overcompensation by showing that imposing balanced coefficients as an optimization constraint does not ensure equalized odds.

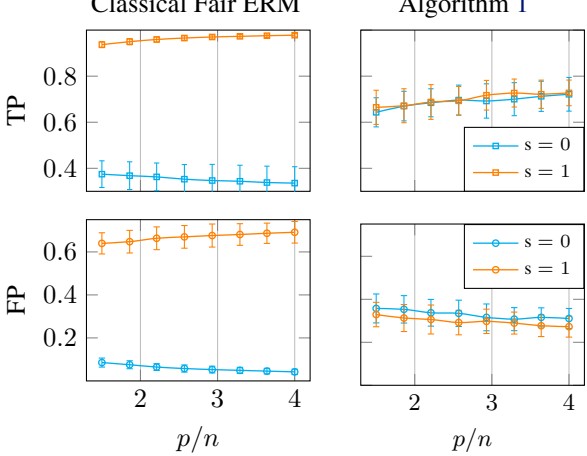

*Figure 4.* Group true positive rates $P(\hat{y} = 1|y = 1, s)$ *(Top)*, and group false positive rates $P(\hat{y} = 1|y = 0, s)$ *(Bottom)*, given by classical fair ERM with $\omega_0 = \omega_1 = 0$ *(Left)*, and the overcompensation method implemented in Algorithm 1 with $\upsilon = 1.5$, $T = 40$ *(Right)*. Classification by logistic regression $\hat{y} = \mathbf{1}_{>0}(f_{\hat{\theta}}(\mathbf{x}))$ with $f_{\hat{\theta}}$ given by Eq. (6) for $\ell = \ell_{LR}$ and $\lambda = 10^{-4}$, on $n = 500$ instances selected randomly from the full training set of size $6 \times 10^4$ Performance evaluated on the full test set of size $10^4$, and averaged over 20 trials with $\pm 1$ standard deviation.

Although the simple data setting of our analysis cannot account for all the complexities of real-world applications, we expect the bias overcompensation strategy to be effective in

a wider range of learning scenarios under overparameterization, as suggested by our experiments on Color-MNIST data. In a broader context, the scheme of tunable bias compensation can help deal with the mismatch between empirical and expected risks, inherent in large models, and problematic to the in-processing bias mitigation approach. It would be interesting to explore theoretically or/and empirically to what extent it improves the accuracy-fairness tradeoff when used alone or combined with pre-processing and post-processing bias mitigation techniques, which can still work in the overparametrized regime.

Our analysis may be extended to the multi-group or continuous case. For the multi-group case, we could consider different patterns, each associated to one group, and the fairness constraint to be applied to all possible pairwise decomposition with bias overcompensation to counter the spurious correlation between the target and sensitive attributes. For a continuous sensitive attribute, since equal average scores imply zero correlation between the target and the sensitive variable in the binary case, we could formulate the fairness penalty as a zero correlation constraint. Then the bias overcompensation would be enforced by imposing a negative correlation. Another direction for extending the analysis is to examine the validity of balanced coefficients as empirical criterion for equalized odds when other data properties, such as group-specific variances, are taken into account and in the negative case what empirical criteria should be used.

## Acknowledgements

This paper has been partially funded by the Agence Nationale de la Recherche under grants ANR-23-CE23-0029 Regul-IA. The authors also acknowledge the support of the AI Cluster ANITI (ANR-23-IACL-0002).

## Impact Statement

This paper presents a theoretical analysis whose goal is to advance the research on the fairness of overparameterized models. There are many potential societal consequences of our work, none of which we feel must be specifically highlighted here.

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

# A. Proof of theoretical results

**Notations.** Here we introduce the asymptotic notation that will be used in Appendix A.3 and the subsequent sections. The big $O$ notation $O(u_n)$ is understood here in probability. When multidimensional objects are concerned, $O(u_n)$ is understood entry-wise. The notation $O_{\|\cdot\|}(\cdot)$ is understood as follows: for a random vector v, $\mathrm{v} = O_{\|\cdot\|}(u_n)$ means that its Euclidean norm is $O(u_n)$ and for a square random matrix $\mathbf{M}$, $\mathbf{M} = O_{\|\cdot\|}(u_n)$ means that the operator norm of $\mathbf{M}$ is $O(u_n)$. The same rules apply to the small $o$ and Big-Theta $\Theta$ notations. Note that under Assumption 4.2 the notations $O(u_n)$ and $O(u_p)$ are interchangeable, since $n, p$ scales linearly. In the following, we will use consistently $O(u_n)$.

## A.1. Proof of Proposition 3.1

Note first that linear independence in the feature space implies that the kernal matrix $K$ is invertible. Recall from Eq. (6) the objective function $q(\boldsymbol{\alpha}, b) := \frac{1}{n} \sum_{i=1}^{n} \ell\left(\frac{1}{n}[\mathbf{K}\boldsymbol{\alpha}]_i + b, \mathrm{y}_i\right) + \frac{\lambda}{2n^2} \boldsymbol{\alpha}^\mathsf{T} \mathbf{K}\boldsymbol{\alpha}$. Its Hessian matrix is given as

$$\mathbf{H} = \begin{bmatrix} \frac{1}{n^3}\mathbf{K}\mathrm{diag}(\mathbf{t})\mathbf{K} + \frac{1}{n^2}\lambda\mathbf{K} & \frac{1}{n^2}\mathbf{K}\mathbf{t} \\ \frac{1}{n^2}\mathbf{t}^\mathsf{T}\mathbf{K} & \frac{1}{n}\mathbf{1}_n^\mathsf{T}\mathbf{t} \end{bmatrix},$$

where $\mathbf{t} := \{\ell''\left(\frac{1}{n}[\mathbf{K}\boldsymbol{\alpha}]_i + b, \mathrm{y}_i\right)\}_{i=1}^{n}$ with $\ell''(r, y) = \frac{\partial^2 \ell(r,y)}{\partial r^2}$. Computing the determinant of $\mathbf{H}$, we get

$$\det \mathbf{H} = \det \left(\frac{1}{n^3}\mathbf{K}\mathrm{diag}(\mathbf{t})\mathbf{K} + \frac{1}{n^2}\lambda\mathbf{K}\right)\left(\frac{1}{n}\mathbf{1}_n^\mathsf{T}\mathbf{t} - \frac{1}{n^4}\mathbf{t}^\mathsf{T}\mathbf{K}\left(\frac{1}{n^3}\mathbf{K}\mathrm{diag}(\mathbf{t})\mathbf{K} + \frac{1}{n^2}\lambda\mathbf{K}\right)^{-1}\mathbf{K}\mathbf{t}\right).$$

Note that $\mathbf{t}$ has positive entries due to the convexity of $\ell$. Therefore, for all $\lambda > 0$, we have

$$\det\left(\frac{1}{n^3}\mathbf{K}\mathrm{diag}(\mathbf{t})\mathbf{K} + \frac{1}{n^2}\lambda\mathbf{K}\right) > 0$$

and

$$\frac{1}{n^4}\mathbf{t}^\mathsf{T}\mathbf{K}\left(\frac{1}{n^3}\mathbf{K}\mathrm{diag}(\mathbf{t})\mathbf{K} + \frac{1}{n^2}\lambda\mathbf{K}\right)^{-1}\mathbf{K}\mathbf{t} < \frac{1}{n^4}\mathbf{t}^\mathsf{T}\mathbf{K}\left(\frac{1}{n^3}\mathbf{K}\mathrm{diag}(\mathbf{t})\mathbf{K}\right)^{-1}\mathbf{K}\mathbf{t} < \frac{1}{n}\mathbf{1}_n^\mathsf{T}\mathbf{t}.$$

We conclude that

$$\det \mathbf{H} > 0.$$

The objective function $q(\boldsymbol{\alpha}, b)$ in Eq. (6) is thus strongly convex. Evidently, the feasible set defined by the two linear constraints in Eq. (6) is a non-empty, convex subset in $\mathbb{R}^{n+1}$. Therefore, the optimization problem of Eq. (6) admits a unique solution $(\hat{\boldsymbol{\alpha}}, \hat{b})$.

## A.2. Proof of Proposition 3.2

Setting $\omega_0, \omega_1 = 0$ in Eq. (4) retrieves the equality constraint

$$\mathbb{E}[f_{\hat{\boldsymbol{\theta}}}(\mathbf{x})|\mathrm{y}, \mathrm{s} = 0] = \mathbb{E}[f_{\hat{\boldsymbol{\theta}}}(\mathbf{x})|\mathrm{y}, \mathrm{s} = 1].$$

It is evident that $\mathbb{E}[f_{\hat{\boldsymbol{\theta}}}(\mathbf{x})|\mathrm{y}, \mathrm{s} = 0] = \mathbb{E}[f_{\hat{\boldsymbol{\theta}}}(\mathbf{x})|\mathrm{y}, \mathrm{s} = 1]$ is a necessary condition for Eq. (2), since the conditional independence between $f_{\hat{\boldsymbol{\theta}}}(\mathbf{x})$ and s given y implies the equality $\mathbb{E}[f_{\hat{\boldsymbol{\theta}}}(\mathbf{x})|\mathrm{y}, \mathrm{s} = 0] = \mathbb{E}[f_{\hat{\boldsymbol{\theta}}}(\mathbf{x})|\mathrm{y}, \mathrm{s} = 1]$. To prove Proposition 3.2, it remains to show that $\mathbb{E}[f_{\hat{\boldsymbol{\theta}}}(\mathbf{x})|\mathrm{y}, \mathrm{s} = 0] = \mathbb{E}[f_{\hat{\boldsymbol{\theta}}}(\mathbf{x})|\mathrm{y}, \mathrm{s} = 1]$ is also a sufficient condition for Eq. (2)

With the feature mapping $\phi(\cdot)$, the classification score $f_{\hat{\boldsymbol{\theta}}}(\mathbf{x}) = \frac{1}{n}\hat{\boldsymbol{\alpha}}^\mathsf{T} k(\mathbf{X}, \mathbf{x}) + \hat{b}$ can be alternatively written as

$$f_{\hat{\boldsymbol{\theta}}}(\mathbf{x}) = \hat{\boldsymbol{\beta}}^\mathsf{T}\phi(\mathbf{x}) + \hat{b}, \quad \text{with } \hat{\boldsymbol{\beta}} = \frac{1}{n}\sum_{i=1}^{n}\hat{\alpha}_i\phi(\mathbf{x}_i).$$

Then, according to the data model in Eq. (7), we have

$$f_{\hat{\boldsymbol{\theta}}}(\mathbf{x}) = \mathrm{y}\hat{\boldsymbol{\beta}}_{\mathcal{S}}^\mathsf{T}\boldsymbol{\mu}_{\mathcal{S}} + \mathrm{s}\hat{\boldsymbol{\beta}}_{\mathcal{S}^c}^\mathsf{T}\boldsymbol{\mu}_{\mathcal{S}^c} + \boldsymbol{\beta}^\mathsf{T}\mathbf{e} + \hat{b}.$$

Since $\mathbf{e}$ is independent of y, s, we note that

$$\mathbb{E}[f_{\hat{\boldsymbol{\theta}}}(\mathbf{x})|\mathrm{y}, \mathrm{s}] = \mathrm{y}\hat{\boldsymbol{\beta}}_{\mathcal{S}}^\mathsf{T}\boldsymbol{\mu}_{\mathcal{S}} + \mathrm{s}\hat{\boldsymbol{\beta}}_{\mathcal{S}^c}^\mathsf{T}\boldsymbol{\mu}_{\mathcal{S}^c} + \hat{b}.$$

Therefore, the equality $\mathbb{E}[f_{\hat{\boldsymbol{\theta}}}(\mathbf{x})|\mathrm{y}, \mathrm{s} = 0] = \mathbb{E}[f_{\hat{\boldsymbol{\theta}}}(\mathbf{x})|\mathrm{y}, \mathrm{s} = 1]$ entails that $\hat{\boldsymbol{\beta}}_{\mathcal{S}^c}^\mathsf{T}\boldsymbol{\mu}_{\mathcal{S}^c} = 0$, in which case $f_{\hat{\boldsymbol{\theta}}}(\mathbf{x})$ is conditionally independent of s given y, thus satisfying Eq. (2). In summary, $\mathbb{E}[f_{\hat{\boldsymbol{\theta}}}(\mathbf{x})|\mathrm{y}, \mathrm{s} = 0] = \mathbb{E}[f_{\hat{\boldsymbol{\theta}}}(\mathbf{x})|\mathrm{y}, \mathrm{s} = 1]$ is a sufficient condition for Eq. (2).

**A.3. Proof of Theorem 4.4**

Let us start by noticing that

$$\frac{1}{n}\mathbf{K}, \left(\frac{1}{n}\mathbf{K}\right)^{-1} = \Theta_{\|\cdot\|}(1), \tag{17}$$

which follows from Assumption 4.1, Assumption 4.2, and standard results in random matrix theory (RMT) (Couillet & Liao, 2022). This ensures

$$\hat{\boldsymbol{\alpha}} = O(1). \tag{18}$$

To see Eq. (18), note first that the empirical risk $\frac{1}{n}\sum_{i=1}^{n}\ell\left(\frac{1}{n}[\mathbf{K}\boldsymbol{\alpha}]_i + b, \mathrm{y}_i\right)$ is minimized at $(\boldsymbol{\alpha}, b) = \left(\left(\frac{1}{n}\mathbf{K}\right)^{-1}\mathbf{y}, 0\right)$, where $\mathbf{y} = \{\mathrm{y}_i\}_{i=1}^{n}$. Since $(\hat{\boldsymbol{\alpha}}, \hat{b})$ is the minimizer of the objective function $q(\boldsymbol{\alpha}, b) := \frac{1}{n}\sum_{i=1}^{n}\ell\left(\frac{1}{n}[\mathbf{K}\boldsymbol{\alpha}]_i + b, \mathrm{y}_i\right) + \frac{\lambda}{2n^2}\boldsymbol{\alpha}^{\mathsf{T}}\mathbf{K}\boldsymbol{\alpha}$, we have $\frac{\lambda}{2n^2}\hat{\boldsymbol{\alpha}}^{\mathsf{T}}\mathbf{K}\hat{\boldsymbol{\alpha}} \leq \frac{\lambda}{2n}\mathbf{y}^{\mathsf{T}}\left(\frac{1}{n}\mathbf{K}\right)^{-1}\mathbf{y} = O(1)$. We thus get Eq. (18) from Eq. (17) and the statistical interchangeability of $\hat{\alpha}_{i\in[n]}$.

According to Proposition 3.1, the optimization problem in Eq. (6) admits a unique solution $(\hat{\boldsymbol{\alpha}}, \hat{b})$. Define the Lagrangian function

$$L(\boldsymbol{\alpha}, b, \psi_0, \psi_1) := q(\boldsymbol{\alpha}, b) + \frac{\psi_0}{n^2}\mathbf{g}_0(\omega_0)^{\mathsf{T}}\mathbf{K}\boldsymbol{\alpha} + \frac{\psi_1}{n^2}\mathbf{g}_1(\omega_1)^{\mathsf{T}}\mathbf{K}\boldsymbol{\alpha},$$

where $q(\boldsymbol{\alpha}, b) := \frac{1}{n}\sum_{i=1}^{n}\ell\left(\frac{1}{n}[\mathbf{K}\boldsymbol{\alpha}]_i + b, \mathrm{y}_i\right) + \frac{\lambda}{2n^2}\boldsymbol{\alpha}^{\mathsf{T}}\mathbf{K}\boldsymbol{\alpha}$ is the objective function, and $\psi_0, \psi_1$ are the Lagrange multipliers for the two linear constraints in Eq. (6), which are rewritten as $\frac{1}{n^2}\mathbf{g}_\mathrm{y}(\omega_\mathrm{y})^{\mathsf{T}}\mathbf{K}\boldsymbol{\alpha} = 0$ with

$$\mathbf{g}_\mathrm{y}(\omega_\mathrm{y}) = n\left(\frac{\omega_\mathrm{y}(\mathbf{q}_{10} + \mathbf{q}_{11})}{n_{10} + n_{11}} - \frac{\omega_\mathrm{y}(\mathbf{q}_{00} + \mathbf{q}_{01})}{n_{00} + n_{01}} + \frac{\mathbf{q}_{\mathrm{y}1}}{n_{\mathrm{y}1}} - \frac{\mathbf{q}_{\mathrm{y}0}}{n_{\mathrm{y}0}}\right).$$

The method of Lagrange multipliers states that the solution $(\hat{\boldsymbol{\alpha}}, \hat{b})$ is found at a stationary point $(\hat{\boldsymbol{\alpha}}, \hat{b}, \hat{\psi}_0, \hat{\psi}_1)$ that cancels all partial derivatives of $L(\boldsymbol{\alpha}, b, \psi_0, \psi_1)$. In other words, we have

$$\frac{\partial L(\hat{\boldsymbol{\alpha}}, \hat{b}, \hat{\psi}_0, \hat{\psi}_1)}{\partial \hat{\boldsymbol{\alpha}}} = -\frac{1}{n^2}\mathbf{K}\hat{\mathbf{a}} + \frac{\lambda}{n^2}\mathbf{K}\hat{\boldsymbol{\alpha}} + \frac{\hat{\psi}_0}{n^2}\mathbf{K}\mathbf{g}_0(\omega_0) + \frac{\hat{\psi}_1}{n^2}\mathbf{K}\mathbf{g}_1(\omega_1) = 0$$

$$\frac{\partial L(\hat{\boldsymbol{\alpha}}, \hat{b}, \hat{\psi}_0, \hat{\psi}_1)}{\partial \hat{b}} = \frac{1}{n}\mathbf{1}_n^{\mathsf{T}}\hat{\mathbf{a}} = 0$$

$$\frac{\partial L(\hat{\boldsymbol{\alpha}}, \hat{b}, \hat{\psi}_0, \hat{\psi}_1)}{\partial \hat{\psi}_0} = \frac{1}{n^2}\mathbf{g}_0(\omega_0)^{\mathsf{T}}\mathbf{K}\hat{\boldsymbol{\alpha}} = 0$$

$$\frac{\partial L(\hat{\boldsymbol{\alpha}}, \hat{b}, \hat{\psi}_0, \hat{\psi}_1)}{\partial \hat{\psi}_1} = \frac{1}{n^2}\mathbf{g}_1(\omega_1)^{\mathsf{T}}\mathbf{K}\hat{\boldsymbol{\alpha}} = 0$$

where we set $\hat{\mathbf{a}} := \left\{-\ell'\left(\frac{1}{n}[\mathbf{K}\hat{\boldsymbol{\alpha}}]_i + \hat{b}, \mathrm{y}_i\right)\right\}_{i=1}^{n}$, and define $\ell'(r, y) = \frac{\partial \ell(r, y)}{\partial r}$. The last two equations ensure that the two linear constraints in Eq. (6) are met for the solution $(\hat{\boldsymbol{\alpha}}, \hat{b})$. The first equation leads to

$$\hat{\boldsymbol{\alpha}} = \frac{1}{\lambda}\hat{\mathbf{a}} - \frac{\hat{\psi}_0}{\lambda}\mathbf{g}_0(\omega_0) - \frac{\hat{\psi}_1}{\lambda}\mathbf{g}_1(\omega_1). \tag{19}$$

Then, using the second equation and the fact that $\mathbf{g}_\mathrm{y}(\omega_\mathrm{y})^{\mathsf{T}}\mathbf{1}_n = 0$, we get

$$\frac{1}{n}\mathbf{1}_n^{\mathsf{T}}\hat{\boldsymbol{\alpha}} = \frac{1}{n}\mathbf{1}_n^{\mathsf{T}}\hat{\mathbf{a}} = 0. \tag{20}$$

Let us define the leave-one-sample-out solution $(\hat{\boldsymbol{\alpha}}_{-i}, \hat{b}_{-i})$ as obtained similarly to $(\hat{\boldsymbol{\alpha}}, \hat{b})$ but without the $i$-th sample:

$$(\hat{\boldsymbol{\alpha}}_{-i}, \hat{b}_{-i}) = \underset{(\boldsymbol{\alpha}_{-i}, b)\in\mathbb{R}^{n-1}\times\mathbb{R}}{\arg\min} \frac{1}{n}\sum_{j\neq i}\ell\left(\frac{1}{n}[\mathbf{K}_{\mathcal{C}_i\mathcal{C}_i}\boldsymbol{\alpha}_{-i}]_j + b, \mathrm{y}_j\right) + \frac{\lambda}{2n^2}\boldsymbol{\alpha}_{-i}^{\mathsf{T}}\mathbf{K}_{\mathcal{C}_i\mathcal{C}_i}\boldsymbol{\alpha}_{-i}$$

$$\text{s.t.} \quad \frac{1}{n^2}[\mathbf{g}_y(\omega_y)]_{\mathcal{C}_i}^{\mathsf{T}}\mathbf{K}_{\mathcal{C}_i\mathcal{C}_i}\boldsymbol{\alpha}_{-i} = 0, \quad \forall y \in \{0, 1\}, \tag{21}$$

where we denote $\mathcal{C}_i = [n] \setminus \{i\}$. Note importantly that $(\hat{\boldsymbol{\alpha}}_{-i}, \hat{b}_{-i})$ is independent of $\mathbf{e}_i$ by construction. Similarly to Eq. (19), we have

$$\hat{\boldsymbol{\alpha}}_{-i} = \frac{1}{\lambda}\hat{\mathbf{a}}_{-i} - \frac{\hat{\psi}_{0(-i)}}{\lambda}[\mathbf{g}_0(\omega_0)]_{\mathcal{C}_i} - \frac{\hat{\psi}_{1(-i)}}{\lambda}[\mathbf{g}_1(\omega_1)]_{\mathcal{C}_i}, \tag{22}$$

where $\hat{\mathbf{a}}_{-i} := \left\{ -\ell'\left( \frac{1}{n}[\mathbf{K}_{\mathcal{C}_i\mathcal{C}_i}\hat{\boldsymbol{\alpha}}_{-i}]_j + \hat{b}_{-i}, \mathbf{y}_j \right) \right\}_{j \neq i}$. Then, using $\frac{1}{n}\mathbf{1}_{n-1}^\mathsf{T}\hat{\mathbf{a}}_{-i} = 0$ and $\frac{1}{n}\mathbf{1}_{n-1}^\mathsf{T}[\mathbf{g}_y(\omega_y)]_{\mathcal{C}_i} = O(n^{-1})$, we obtain

$$\frac{1}{n}\mathbf{1}_{n-1}^\mathsf{T}\hat{\boldsymbol{\alpha}}_{-i} = O(n^{-1}). \tag{23}$$

Since $\ell(\cdot, y)$ is three-times differentiable, taking the difference Eq. (19) - Eq. (22), we get

$$\hat{\boldsymbol{\alpha}}_{\mathcal{C}_i} - \hat{\boldsymbol{\alpha}}_{-i} = -\frac{1}{\lambda n}\text{diag}(\hat{\mathbf{t}}_{-i})\mathbf{K}_{\mathcal{C}_i\mathcal{C}_i}\left(\hat{\boldsymbol{\alpha}}_{\mathcal{C}_i} - \hat{\boldsymbol{\alpha}}_{-i}\right) - \frac{1}{\lambda n}\text{diag}(\hat{\mathbf{t}}_{-i})\mathbf{K}_{\mathcal{C}_i\{i\}}\hat{\alpha}_i - (\hat{b} - \hat{b}_{-i})\hat{\mathbf{t}}_{-i}$$
$$- \frac{(\hat{\psi}_0 - \hat{\psi}_{0(-i)})}{n\lambda}[\mathbf{g}_0(\omega_0)]_{\mathcal{C}_i} - \frac{(\hat{\psi}_1 - \hat{\psi}_{1(-i)})}{n\lambda}[\mathbf{g}_1(\omega_1)]_{\mathcal{C}_i},$$

for some $\hat{\mathbf{t}}_{-i} = O(1)$. We thus obtain that

$$\hat{\boldsymbol{\alpha}}_{\mathcal{C}_i} - \hat{\boldsymbol{\alpha}}_{-i} = \left( \boldsymbol{I}_{n-1} + \frac{1}{\lambda n}\text{diag}(\hat{\mathbf{t}}_{-i})\mathbf{K}_{\mathcal{C}_i\mathcal{C}_i} \right)^{-1} \left[ -\frac{1}{\lambda n}\text{diag}(\hat{\mathbf{t}}_{-i})\mathbf{K}_{\mathcal{C}_i\{i\}}\hat{\alpha}_i - (\hat{b} - \hat{b}_{-i})\hat{\mathbf{t}}_{-i} \right.$$
$$\left. - \frac{(\hat{\psi}_0 - \hat{\psi}_{0(-i)})}{\lambda}[\mathbf{g}_0(\omega_0)]_{\mathcal{C}_i} - \frac{(\hat{\psi}_1 - \hat{\psi}_{1(-i)})}{\lambda}[\mathbf{g}_1(\omega_1)]_{\mathcal{C}_i} \right]$$

It follows from the above expression of $\hat{\boldsymbol{\alpha}}_{\mathcal{C}_i} - \hat{\boldsymbol{\alpha}}_{-i}$, also the linear constraints in both Eq. (4) and Eq. (21) that

$$\frac{1}{n^2}[\mathbf{g}_y(\omega_y)]_{\mathcal{C}_i}^\mathsf{T}\mathbf{K}_{\mathcal{C}_i\mathcal{C}_i}(\hat{\boldsymbol{\alpha}}_{\mathcal{C}_i} - \hat{\boldsymbol{\alpha}}_{-i}) = \frac{1}{n}[\mathbf{g}_y(\omega_y)]_{\mathcal{C}_i}^\mathsf{T}\mathbf{K}_{\mathcal{C}_i\mathcal{C}_i}\left( \boldsymbol{I}_{n-1} + \frac{1}{\lambda n}\text{diag}(\hat{\mathbf{t}}_{-i})\mathbf{K}_{\mathcal{C}_i\mathcal{C}_i} \right)^{-1}\left[ -(\hat{b} - \hat{b}_{-i})\hat{\mathbf{t}}_{-i} \right.$$
$$\left. - \frac{(\hat{\psi}_0 - \hat{\psi}_{0(-i)})}{\lambda}[\mathbf{g}_0(\omega_0)]_{\mathcal{C}_i} - \frac{(\hat{\psi}_1 - \hat{\psi}_{1(-i)})}{\lambda}[\mathbf{g}_1(\omega_1)]_{\mathcal{C}_i} \right] + O(n^{-1})$$
$$= O(n^{-1}).$$

Similarly, as a result of Eq. (20) and Eq. (23), we have

$$\frac{1}{n}\mathbf{1}_{n-1}^\mathsf{T}(\hat{\boldsymbol{\alpha}}_{\mathcal{C}_i} - \hat{\boldsymbol{\alpha}}_{-i}) = \frac{1}{n}\mathbf{1}_{n-1}^\mathsf{T}\left( \boldsymbol{I}_{n-1} + \frac{1}{\lambda n}\text{diag}(\hat{\mathbf{t}}_{-i})\mathbf{K}_{\mathcal{C}_i\mathcal{C}_i} \right)^{-1}\left[ -(\hat{b} - \hat{b}_{-i})\hat{\mathbf{t}}_{-i} \right.$$
$$\left. - \frac{(\hat{\psi}_0 - \hat{\psi}_{0(-i)})}{\lambda}[\mathbf{g}_0(\omega_0)]_{\mathcal{C}_i} - \frac{(\hat{\psi}_1 - \hat{\psi}_{1(-i)})}{\lambda}[\mathbf{g}_1(\omega_1)]_{\mathcal{C}_i} \right] + O(n^{-1})$$
$$= O(n^{-1}).$$

Note that the entries of $\hat{\mathbf{t}}_{-i}$ are strictly positive due to the strong convexity of $\ell(\cdot, y)$, which entails the asymptotic linear independence of $\hat{\mathbf{t}}_{-i}, [\mathbf{g}_0(\omega_0)]_{\mathcal{C}_i}, [\mathbf{g}_1(\omega_1)]_{\mathcal{C}_i}$, and that of $\mathbf{1}_{n-1}, \frac{1}{n}[\mathbf{g}_0(\omega_0)]_{\mathcal{C}_i}^\mathsf{T}\mathbf{K}_{\mathcal{C}_i\mathcal{C}_i}, \frac{1}{n^2}[\mathbf{g}_1(\omega_1)]_{\mathcal{C}_i}^\mathsf{T}\mathbf{K}_{\mathcal{C}_i\mathcal{C}_i}$. Consequently, the last two approximation in separate lines entail that $\hat{\psi}_0 - \hat{\psi}_{0(-i)} = O(n^{-1})$, $\hat{\psi}_1 - \hat{\psi}_{1(-i)} = O(n^{-1})$, and $\hat{b} - \hat{b}_{-i} = O(n^{-1})$. In summary, we have

$$\hat{\boldsymbol{\alpha}}_{\mathcal{C}_i} - \hat{\boldsymbol{\alpha}}_{-i} = O(n^{-\frac{1}{2}}) \tag{24}$$

With a slight abuse of notation, let us set from now on

$$\hat{\mathbf{t}}_{-i} := \{\ell''\left( \frac{1}{n}[\mathbf{K}_{\mathcal{C}_i\mathcal{C}_i}\hat{\boldsymbol{\alpha}}_{-i}]_j + \hat{b}_{-i}, \mathbf{y}_j \right)\}_{j \neq i}.$$

Finally, we get

$$\hat{\boldsymbol{\alpha}}_{\mathcal{C}_i} - \hat{\boldsymbol{\alpha}}_{-i} = -\left( \boldsymbol{I}_{n-1} + \frac{1}{\lambda n}\text{diag}(\hat{\mathbf{t}}_{-i})\mathbf{K}_{\mathcal{C}_i\mathcal{C}_i} \right)^{-1}\frac{1}{\lambda n}\text{diag}(\hat{\mathbf{t}}_{-i})\mathbf{K}_{\mathcal{C}_i\{i\}}\hat{\alpha}_i + O(n^{-1}). \tag{25}$$

Denote $\mathbf{z}_i := \phi(\mathbf{x}_i)$, $\mathbf{Z} := [\mathbf{z}_1, \ldots, \mathbf{z}_n] \in \mathbb{R}^{p \times n}$, $\mathbf{Z}_{-i} := [\mathbf{z}_1, \ldots, \mathbf{z}_{i-1}, \mathbf{z}_{i+1}, \mathbf{z}_n] \in \mathbb{R}^{p \times (n-1)}$. And define $\hat{\boldsymbol{\beta}} := \frac{1}{n} \mathbf{Z} \hat{\boldsymbol{\alpha}}$, $\hat{\boldsymbol{\beta}}_{-i} := \frac{1}{n} \mathbf{Z}_{-i} \hat{\boldsymbol{\alpha}}_{-i}$, which allows us to write

$$\frac{1}{n}[\mathbf{K}\hat{\boldsymbol{\alpha}}]_i = \hat{\boldsymbol{\beta}}^{\mathsf{T}} \mathbf{z}_i, \quad \frac{1}{n}[\mathbf{K}_{\mathcal{C}_i \mathcal{C}_i} \hat{\boldsymbol{\alpha}}_{-i}]_i = \hat{\boldsymbol{\beta}}_{-i}^{\mathsf{T}} \mathbf{z}_i.$$

The key difference between $\hat{\boldsymbol{\beta}}$ and $\hat{\boldsymbol{\beta}}_{-i}$ is that $\hat{\boldsymbol{\beta}}_{-i}$ is independent of $\phi(\mathbf{x}_i)$. And $\hat{\boldsymbol{\beta}}$ is close to $\hat{\boldsymbol{\beta}}_{-i}$, as

$$\hat{\boldsymbol{\beta}} - \hat{\boldsymbol{\beta}}_{-i} = \frac{1}{n} \left( \boldsymbol{I}_p + \frac{1}{\lambda n} \mathbf{Z}_{-i} \mathrm{diag}(\hat{\mathbf{t}}_{-i}) \mathbf{Z}_{-i}^{\mathsf{T}} \right)^{-1} \hat{\alpha}_i \mathbf{z}_i + O_{\|\cdot\|}(n^{-1}) = O_{\|\cdot\|}(n^{-\frac{1}{2}}), \tag{26}$$

which follows from Eq. (25). Then, we obtain

$$\hat{\boldsymbol{\beta}}^{\mathsf{T}} \mathbf{z}_i - \hat{\boldsymbol{\beta}}_{-i}^{\mathsf{T}} \mathbf{z}_i = \frac{\|\mathbf{z}_i\|^2}{n} \hat{\alpha}_i + \frac{1}{n} \mathbf{z}_i^{\mathsf{T}} \mathbf{Z}_{-i} \left( \hat{\boldsymbol{\alpha}}_{\mathcal{C}_i} - \hat{\boldsymbol{\alpha}}_{-i} \right) = \kappa_i \hat{\alpha}_i + O(n^{-\frac{1}{2}}),$$

where

$$\kappa_i = \frac{1}{n} \mathbf{z}_i^{\mathsf{T}} \left( \boldsymbol{I}_p + \frac{1}{\lambda n} \mathbf{Z}_{-i} \mathrm{diag}(\hat{\mathbf{t}}_{-i}) \mathbf{Z}_{-i}^{\mathsf{T}} \right)^{-1} \mathbf{z}_i.$$

Let us define

$$\kappa := \frac{1}{n} \mathrm{Tr}\, \boldsymbol{\Sigma} \left( \boldsymbol{I}_p + \frac{1}{\lambda n} \mathbf{Z} \mathrm{diag}(\hat{\mathbf{t}}) \mathbf{Z}^{\mathsf{T}} \right)^{-1} \tag{27}$$

with

$$\hat{\mathbf{t}} := \{\ell'' \left( \frac{1}{n}[\mathbf{K}\hat{\boldsymbol{\alpha}}]_i + \hat{b}, \mathbf{y}_i \right)\}_{i=1}^n.$$

Since $\hat{\boldsymbol{\alpha}}_{\mathcal{C}_i} - \hat{\boldsymbol{\alpha}}_{-i} = O(n^{-\frac{1}{2}})$ and $\hat{b} - \hat{b}_{-i} = O(n^{-1})$, we have

$$\hat{\mathbf{t}}_{\mathcal{C}_i} - \hat{\mathbf{t}}_{-i} = O(n^{-\frac{1}{2}}).$$

Consequently,

$$\kappa = \frac{1}{n} \mathrm{Tr}\, \boldsymbol{\Sigma} \left( \boldsymbol{I}_p + \frac{1}{\lambda n} \mathbf{Z}_{-i} \mathrm{diag}(\hat{\mathbf{t}}_{-i}) \mathbf{Z}_{-i}^{\mathsf{T}} \right)^{-1} + O(n^{-\frac{1}{2}}) = \kappa_i + O(n^{-\frac{1}{2}}).$$

In the end, we obtain that

$$\hat{\boldsymbol{\beta}}^{\mathsf{T}} \mathbf{z}_i - \hat{\boldsymbol{\beta}}_{-i}^{\mathsf{T}} \mathbf{z}_i = \kappa \hat{\alpha}_i + O(n^{-\frac{1}{2}}), \tag{28}$$

and

$$\hat{\boldsymbol{\beta}}^{\mathsf{T}} \mathbf{z}_j - \hat{\boldsymbol{\beta}}_{-i}^{\mathsf{T}} \mathbf{z}_j = O(n^{-\frac{1}{2}}), \quad \forall j \neq i. \tag{29}$$

From Eq. (28), it follows that

$$\widehat{\mathbb{E}}[h(\hat{\boldsymbol{\beta}}^{\mathsf{T}} \mathbf{z}_i - \kappa \hat{\alpha})|\mathbf{y}, \mathbf{s}] = \widehat{\mathbb{E}}[h(\hat{\boldsymbol{\beta}}_{-i}^{\mathsf{T}} \mathbf{z}_i)|\mathbf{y}, \mathbf{s}] + O(n^{-\frac{1}{2}}),$$

for any bounded Lipschitz $h$. Since

$$\widehat{\mathbb{E}}[h(\hat{\boldsymbol{\beta}}_{-i}^{\mathsf{T}} \mathbf{z}_i)|\mathbf{y}, \mathbf{s}] = \frac{1}{n_{\mathrm{ys}}} \sum_{i \in \mathcal{I}_{\mathrm{ys}}} h(\hat{\boldsymbol{\beta}}_{-i}^{\mathsf{T}} \mathbf{z}_i)$$

with $\mathcal{I}_{\mathrm{ys}} := \{i \in [n] | \mathbf{y}_i = \mathbf{y}, \mathbf{s}_i = \mathbf{s}\}$, we have

$$\mathrm{Var}\left( \widehat{\mathbb{E}}[h(\hat{\boldsymbol{\beta}}_{-i}^{\mathsf{T}} \mathbf{z}_i)|\mathbf{y}, \mathbf{s}] \right) = \frac{1}{n_{\mathrm{ys}}^2} \sum_{i \in \mathcal{I}_{\mathrm{ys}}} \mathrm{Var}\left( h(\hat{\boldsymbol{\beta}}_{-i}^{\mathsf{T}} \mathbf{z}_i) | \mathbf{y}_i, \mathbf{s}_i \right) + \frac{1}{n_{\mathrm{ys}}^2} \sum_{i \neq j \in \mathcal{I}_{\mathrm{ys}}} \mathrm{Cov}\left( h(\hat{\boldsymbol{\beta}}_{-i}^{\mathsf{T}} \mathbf{z}_i), h(\hat{\boldsymbol{\beta}}_{-j}^{\mathsf{T}} \mathbf{z}_j) | \mathbf{y}_i, \mathbf{s}_i, \mathbf{y}_j, \mathbf{s}_j \right).$$

Let $\hat{\boldsymbol{\beta}}_{-ij}$ be the leave-two-samples-out solution understood similarly to $\hat{\boldsymbol{\beta}}_{-i}$, but obtained by removing the $i$-th and the $j$-th samples, therefore independent of $\mathbf{e}_i, \mathbf{e}_j$ by construction. Applying Eq. (29) w.r.t. $\hat{\boldsymbol{\beta}}_{-ij}$ leads to

$$\mathrm{Cov}\left( h(\hat{\boldsymbol{\beta}}_{-i}^{\mathsf{T}} \mathbf{z}_i), h(\hat{\boldsymbol{\beta}}_{-j}^{\mathsf{T}} \mathbf{z}_j) | \mathbf{y}_i, \mathbf{s}_i, \mathbf{y}_j, \mathbf{s}_j \right) = \mathrm{Cov}\left( h(\hat{\boldsymbol{\beta}}_{-ij}^{\mathsf{T}} \mathbf{z}_i), h(\hat{\boldsymbol{\beta}}_{-ij}^{\mathsf{T}} \mathbf{z}_j) | \mathbf{y}_i, \mathbf{s}_i, \mathbf{y}_j, \mathbf{s}_j \right) + O(n^{-\frac{1}{2}})$$

Note that
$$\mathrm{Cov}\big(h(\hat{\boldsymbol{\beta}}_{-ij}^{\mathsf{T}}\mathbf{z}_i), h(\hat{\boldsymbol{\beta}}_{-ij}^{\mathsf{T}}\mathbf{z}_j)\big|\mathbf{y}_i, \mathbf{s}_i, \mathbf{y}_j, \mathbf{s}_j\big) = 0,$$

due to the fact that $\hat{\boldsymbol{\beta}}_{-ij}^{\mathsf{T}}\mathbf{z}_i$ and $\hat{\boldsymbol{\beta}}_{-ij}^{\mathsf{T}}\mathbf{z}_j$) are independent when conditioned on $\hat{\boldsymbol{\beta}}_{-ij}$. Thus, we get

$$\mathrm{Var}\left(\widehat{\mathbb{E}}[h(\hat{\boldsymbol{\beta}}_{-i}^{\mathsf{T}}\mathbf{z}_i)|\mathbf{y},\mathbf{s}]\right) = O(n^{-\frac{1}{2}}),$$

which entails

$$\widehat{\mathbb{E}}[h(\hat{\boldsymbol{\beta}}_{-i}^{\mathsf{T}}\mathbf{z}_i)|\mathbf{y},\mathbf{s}] = \mathbb{E}[h(\hat{\boldsymbol{\beta}}_{-i}^{\mathsf{T}}\mathbf{z}_i)|\mathbf{y}_i=\mathbf{y}, \mathbf{s}_i=\mathbf{s}] + O(n^{-\frac{1}{4}}) = \mathbb{E}[h(\hat{\boldsymbol{\beta}}^{\mathsf{T}}\mathbf{z})|\mathbf{y},\mathbf{s}] + O(n^{-\frac{1}{4}}),$$

where the second equality is justified by Eq. (26). In summary, we have

$$\widehat{\mathbb{E}}[h(\hat{\boldsymbol{\beta}}^{\mathsf{T}}\mathbf{z}_i - \kappa\hat{\alpha})|\mathbf{y},\mathbf{s}] - \mathbb{E}[h(\hat{\boldsymbol{\beta}}^{\mathsf{T}}\mathbf{z})|\mathbf{y},\mathbf{s}] \to 0. \tag{30}$$

Now, to prove Eq. (10), it remains to demonstrate the concentration of $\hat{b}$.

Recall from Eq. (19) that
$$\hat{\alpha}_i = -\frac{1}{\lambda}\ell'(\hat{\boldsymbol{\beta}}^{\mathsf{T}}\mathbf{z}_i + \hat{b}, y_i) + c_{\mathbf{y}_i\mathbf{s}_i}$$

with
$$c_{\mathrm{ys}} = (1 - 2\mathrm{s})\left(\frac{\hat{\psi}_0}{\lambda}\frac{n}{n_{0\mathrm{s}}} + \frac{\hat{\psi}_1}{\lambda}\frac{n}{n_{1\mathrm{s}}}\right) + (1 - 2\mathrm{y})\frac{\hat{\psi}_0\omega_0 + \hat{\psi}_1\omega_1}{\lambda}\frac{n}{n_{\mathrm{y}0} + n_{\mathrm{y}_1}}.$$

Then we get from Eq. (28) that

$$\hat{\boldsymbol{\beta}}_{-i}^{\mathsf{T}}\mathbf{z}_i + \hat{b} + \kappa c_{\mathbf{y}_i\mathbf{s}_i} = \hat{\boldsymbol{\beta}}^{\mathsf{T}}\mathbf{z}_i\hat{b} + \frac{\kappa}{\lambda}\ell'(\hat{\boldsymbol{\beta}}^{\mathsf{T}}\mathbf{z}_i + \hat{b}, y_i) + O(n^{-\frac{1}{2}}).$$

Define the proximal operator as
$$\mathrm{prox}_{f,\tau}(x) = \arg\min_{t\in\mathbb{R}} f(t) + \frac{\tau}{2}\|t - x\|^2.$$

We have
$$\hat{\boldsymbol{\beta}}^{\mathsf{T}}\mathbf{z}_i + \hat{b} = \mathrm{prox}_{\ell(\cdot,\mathbf{y}_i),\kappa/\lambda}(\hat{\boldsymbol{\beta}}_{-i}^{\mathsf{T}}\mathbf{z}_i + \hat{b} + \kappa c_{\mathbf{y}_i\mathbf{s}_i}) + O(n^{-\frac{1}{2}}),$$

Define the mappings $u_{ys}(x) = \frac{1}{\kappa}\left(\mathrm{prox}_{\ell(\cdot,y),\kappa/\lambda}(x + \kappa c_{ys}) - x\right)$ for $y, s \in \{0, 1\}$, which allows us to rewrite Eq. (28) as

$$\hat{\alpha}_i = u_{\mathbf{y}_i\mathbf{s}_i}(\hat{\boldsymbol{\beta}}_{-i}^{\mathsf{T}}\mathbf{z}_i + \hat{b}) + O(n^{-\frac{1}{2}}). \tag{31}$$

Using the concentration result in Eq. (30), we have

$$\frac{1}{n}\sum_{i=1}^{n}\hat{\alpha}_i - \sum_{y,s=0}^{1}\frac{n_{ys}}{n}\mathbb{E}[u_{ys}(\hat{\boldsymbol{\beta}}^{\mathsf{T}}\mathbf{z} + \hat{b})|\mathbf{y} = y, \mathbf{s} = s, \hat{b}] \to 0$$

Since $\frac{1}{n}\sum_{i=1}^{n}\hat{\alpha}_i = 0$ is imposed by the optimization solution as shown in Eq. (20), the above convergence implies

$$\hat{b} - \tilde{b} \to 0, \tag{32}$$

where $\tilde{b}$ is a deterministic value given by

$$P(\mathbf{y}_i = y, \mathbf{s}_i = s)\mathbb{E}[u_{ys}(\hat{\boldsymbol{\beta}}^{\mathsf{T}}\mathbf{z} + \tilde{b})|\mathbf{y} = y, \mathbf{s} = s] = 0.$$

In the end, we have
$$\widehat{\mathbb{E}}[h(\hat{\boldsymbol{\beta}}^{\mathsf{T}}\mathbf{z}_i + \hat{b} - \kappa\hat{\alpha})|\mathbf{y},\mathbf{s}] - \mathbb{E}[h(\hat{\boldsymbol{\beta}}^{\mathsf{T}}\mathbf{z} + \hat{b})|\mathbf{y},\mathbf{s}] \to 0, \tag{33}$$

which proves Eq. (10).

We proceed to demonstrate Eq. (9) by characterizing the asymptotic distribution of $\hat{\boldsymbol{\beta}}^{\mathsf{T}}\mathbf{z}$.

Remark, as a consequence of Eq. (17) and Eq. (18), that

$$\hat{\boldsymbol{\beta}} = \frac{1}{n}\mathbf{K}\hat{\boldsymbol{\alpha}} = O_{\|\cdot\|}(1).$$

Since the independent features have comparable conditional means w.r.t. $(\mathbf{y}, \mathbf{s})$ and commensurable variances according to Assumption 4.2, $\hat{\boldsymbol{\beta}}$ has also comparable entries, which implies

$$\hat{\boldsymbol{\beta}} = O(n^{-\frac{1}{2}}). \tag{34}$$

From Eq. (34), it follows that

$$\hat{\alpha}_i = u_{\mathbf{y}_i\mathbf{s}_i}\left(\sum_{k\neq l}[\hat{\boldsymbol{\beta}}_{-i}]_k[\mathbf{z}_i]_k + \hat{b}\right) + u'_{\mathbf{y}_i\mathbf{s}_i}\left(\sum_{k\neq l}[\hat{\boldsymbol{\beta}}_{-i}]_k[\mathbf{z}_i]_k + \hat{b}\right)[\hat{\boldsymbol{\beta}}_{-i}]_l[\mathbf{z}_i]_l + O(n^{-1}).$$

Since $\hat{\boldsymbol{\beta}} = \frac{1}{n}\sum_{i=1}^n \hat{\alpha}_i\mathbf{z}_i$, using the above approximation of $\hat{\alpha}_i$, we have

$$[\hat{\boldsymbol{\beta}}]_l = \frac{1}{n}\sum_{i=1}^n \hat{\alpha}_i[\mathbf{z}_i]_l = \frac{1}{n}\sum_{i=1}^n u_{\mathbf{y}_i\mathbf{s}_i}\left(\sum_{k\neq l}[\hat{\boldsymbol{\beta}}_{-i}]_k[\mathbf{z}_i]_k + \hat{b}\right)[\mathbf{z}_i]_l + \frac{1}{n}\sum_{i=1}^n u'_{\mathbf{y}_i\mathbf{s}_i}\left(\sum_{k\neq l}[\hat{\boldsymbol{\beta}}_{-i}]_k[\mathbf{z}_i]_k + \hat{b}\right)[\hat{\boldsymbol{\beta}}_{-i}]_l[\mathbf{z}_i]_l^2 + O(n^{-1}). \tag{35}$$

Applying Eq. (33), we get

$$\frac{1}{n}\sum_{i=1}^n u'_{\mathbf{y}_i\mathbf{s}_i}\left(\sum_{k\neq l}[\hat{\boldsymbol{\beta}}_{-i}]_k[\mathbf{z}_i]_k + \hat{b}\right)[\mathbf{z}_i]_l^2 = \sum_{y,s=0}^1 \frac{n_{ys}}{n}\mathbb{E}\left[u'_{ys}\left(\sum_{k\neq l}[\hat{\boldsymbol{\beta}}]_k[\mathbf{z}]_k + \hat{b}\right)[\mathbf{z}]_l^2\bigg|\mathbf{y}=y,\mathbf{s}=s\right] + o(1)$$

$$= \sum_{y,s=0}^1 \frac{n_{ys}}{n}\mathbb{E}\left[u'_{ys}\left(\sum_{k\neq l}[\hat{\boldsymbol{\beta}}]_k[\mathbf{z}]_k + \hat{b}\right)\bigg|\mathbf{y}=y,\mathbf{s}=s\right]\mathbb{E}[[\mathbf{z}]_l^2|\mathbf{y}=y,\mathbf{s}=s] + o(1)$$

$$= \sigma_l \sum_{y,s=0}^1 \frac{n_{ys}}{n}\mathbb{E}\left[u'_{ys}(\hat{\boldsymbol{\beta}}^{\mathsf{T}}\mathbf{z} + \hat{b})\big|\mathbf{y}=y,\mathbf{s}=s\right] + o(1).$$

Set

$$\nu = -\sum_{y,s=0}^1 \frac{n_{ys}}{n}\mathbb{E}\left[u'_{ys}(\hat{\boldsymbol{\beta}}^{\mathsf{T}}\mathbf{z} + \hat{b})\big|\mathbf{y}=y,\mathbf{s}=s\right],$$

we have from Eq. (35) that

$$(1+\nu\sigma_l)[\hat{\boldsymbol{\beta}}]_l = \frac{1}{n}\sum_{i=1}^n u_{\mathbf{y}_i\mathbf{s}_i}\left(\sum_{k\neq l}[\hat{\boldsymbol{\beta}}_{-i}]_k[\mathbf{z}_i]_k + \hat{b}\right)[\mathbf{z}_i]_l + o(n^{-\frac{1}{2}}),.$$

In summary,

$$(1+\nu\sigma_l)[\hat{\boldsymbol{\beta}}]_l = \widehat{P}(\mathbf{y}=1)\widehat{\mathbb{E}}[\hat{\alpha}|\mathbf{y}=1][\boldsymbol{\mu}]_l + \mathbf{w}_l + o(n^{-\frac{1}{2}}), \quad \forall l \in \mathcal{S},$$
$$(1+\nu\sigma_l)[\hat{\boldsymbol{\beta}}]_l = \widehat{P}(\mathbf{s}=1)\widehat{\mathbb{E}}[\hat{\alpha}|\mathbf{s}=1][\boldsymbol{\mu}]_l + \mathbf{w}_l + o(n^{-\frac{1}{2}}), \quad \forall l \in \mathcal{S}^{\mathrm{c}}, \tag{36}$$

where $\mathbf{w}_l := \frac{1}{n}\sum_{i=1}^n u_{\mathbf{y}_i\mathbf{s}_i}\left(\sum_{k\neq l}[\hat{\boldsymbol{\beta}}_{-i}]_k[\mathbf{z}_i]_k + \hat{b}\right)[\mathbf{e}_i]_l$.

Since $\frac{1}{n}\mathbf{1}_n^{\mathsf{T}}\hat{\alpha} = 0$, we notice that

$$\widehat{P}(\mathbf{y}=1)\widehat{\mathbb{E}}[\hat{\alpha}|\mathbf{y}=1] = \widehat{P}(\mathbf{y}=0)\widehat{P}(\mathbf{y}=1)\Delta\widehat{\mathbb{E}}[\hat{\alpha}|\mathbf{y}]$$
$$\widehat{P}(\mathbf{s}=1)\widehat{\mathbb{E}}[\hat{\alpha}|\mathbf{s}=1] = \widehat{P}(\mathbf{s}=0)\widehat{P}(\mathbf{s}=1)\Delta\widehat{\mathbb{E}}[\hat{\alpha}|\mathbf{s}],$$

which follow from $\widehat{P}(\mathrm{y}=0)\widehat{\mathbb{E}}[\hat{\alpha}|\mathrm{y}=0] + \widehat{P}(\mathrm{y}=1)\widehat{\mathbb{E}}[\hat{\alpha}|\mathrm{y}=1] = 0$ and $\widehat{P}(\mathrm{s}=0)\widehat{\mathbb{E}}[\hat{\alpha}|\mathrm{s}=0] + \widehat{P}(\mathrm{s}=1)\widehat{\mathbb{E}}[\hat{\alpha}|\mathrm{s}=1] = 0$. We can thus rewrite Eq. (36) as

$$\hat{\boldsymbol{\beta}} = \mathbf{D}\tilde{\boldsymbol{\beta}} + \mathbf{Q}(\nu)\mathbf{w} + o_{\|\cdot\|}(1), \tag{37}$$

where $\mathbf{D} = \mathrm{diag}\left(\left[\Delta\widehat{\mathbb{E}}[\hat{\alpha}|\mathrm{y}]\mathbf{1}_{|\mathcal{S}|}^{\mathsf{T}} \quad \Delta\widehat{\mathbb{E}}[\hat{\alpha}|\mathrm{s}]\mathbf{1}_{|\mathcal{S}^c|}^{\mathsf{T}}\right]\right), \tilde{\boldsymbol{\beta}} = \mathbf{Q}(\nu)\begin{bmatrix}\prod_{\mathrm{y}=0}^{1}\widehat{P}(\mathrm{y}=y)\boldsymbol{\mu}_{\mathcal{S}} \\ \prod_{\mathrm{s}=0}^{1}\widehat{P}(\mathrm{s}=s)\boldsymbol{\mu}_{\mathcal{S}^c}\end{bmatrix}$ and $\mathbf{Q}(\nu) := (\boldsymbol{I}_p + \nu\boldsymbol{\Sigma})^{-1}$. Therefore, we obtain

$$\hat{\boldsymbol{\beta}}^{\mathsf{T}}\mathbf{z} = \eta_{\mathrm{ys}} + \mathbf{D}\tilde{\boldsymbol{\beta}}^{\mathsf{T}}\mathbf{e} + \mathbf{z}^{\mathsf{T}}\mathbf{Q}(\nu)\mathbf{w} + o(1) \tag{38}$$

where $\eta_{\mathrm{ys}}$ is as given in Theorem 4.4. It follows from Eq. (33) and Eq. (31) that

$$\mathrm{Var}(\mathbf{D}\tilde{\boldsymbol{\beta}}^{\mathsf{T}}\mathbf{e}) = \tilde{\boldsymbol{\beta}}_{\mathcal{S}}^{\mathsf{T}}\boldsymbol{\Sigma}_{\mathcal{S}}\tilde{\boldsymbol{\beta}}_{\mathcal{S}}\Delta\widehat{\mathbb{E}}[\hat{\alpha}|\mathrm{y}]^2 + \tilde{\boldsymbol{\beta}}_{\mathcal{S}^c}^{\mathsf{T}}\boldsymbol{\Sigma}_{\mathcal{S}^c}\tilde{\boldsymbol{\beta}}_{\mathcal{S}^c}\Delta\widehat{\mathbb{E}}[\hat{\alpha}|\mathrm{s}]^2 + o(1).$$

Now we move on to study the asymptotic behavior of $\mathbf{w}$, which characterizes the noise term $\mathbf{z}^{\mathsf{T}}\mathbf{Q}(\nu)\mathbf{w}$ in Eq. (20). Indeed, since

$$\mathbb{E}[\mathbf{w}_l] = 0, \quad \forall l \in [p],$$

we have

$$\mathbb{E}\mathbf{z}^{\mathsf{T}}\mathbf{Q}(\nu)\mathbf{w}|\mathrm{y},\mathrm{s}] = 0$$

Therefore, it suffices to characterize the variance of $\mathbf{z}^{\mathsf{T}}\mathbf{Q}(\nu)\mathbf{w}$.

Let $\hat{\boldsymbol{\beta}}_{-i}^{[jl]}$ be the solution obtained without the $i$-th sample and with $[\mathbf{e}_j]_l$ set to zero. With approximation arguments similar to those leading to Eq. (26), we have

$$\hat{\boldsymbol{\beta}}_{-i} - \hat{\boldsymbol{\beta}}_{-i}^{[jl]} = O_{\|\cdot\|}(n^{-1}).$$

and

$$\sum_{k\neq l}[\hat{\boldsymbol{\beta}}_{-i}]_k[\mathbf{z}_i]_k - \sum_{k\neq l}[\hat{\boldsymbol{\beta}}_{-i}^{[jl]}]_k[\mathbf{z}_i]_k = \frac{r}{n}[\mathbf{e}_j]_l + o(n^{-1})$$

for some $r = O(1)$, which leads to

$$u_{\mathrm{y}_i\mathrm{s}_i}\left(\sum_{k\neq l}[\hat{\boldsymbol{\beta}}_{-i}]_k[\mathbf{z}_i]_k + \hat{b}\right) = u_{\mathrm{y}_i\mathrm{s}_i}\left(\sum_{k\neq l}[\hat{\boldsymbol{\beta}}_{-i}^{[jl]}]_k[\mathbf{z}_i]_k + \hat{b}\right) + u'_{\mathrm{y}_i\mathrm{s}_i}\left(\sum_{k\neq l}[\hat{\boldsymbol{\beta}}_{-i}^{[jl]}]_k[\mathbf{z}_i]_k + \hat{b}\right)\frac{r}{n}[\mathbf{e}_j]_l + o(n^{-1}).$$

Since $(\sum_{k\neq l}[\hat{\boldsymbol{\beta}}_{-i}^{[jl]}]_k[\mathbf{z}_i]_k$ is independent of both $[\mathbf{e}_i]_l$ and $[\mathbf{e}_j]_l$, we note that

$$\mathrm{Cov}\left(u_{\mathrm{y}_i\mathrm{s}_i}\left(\sum_{k\neq l}[\hat{\boldsymbol{\beta}}_{-i}]_k[\mathbf{z}_i]_k + \hat{b}\right)[\mathbf{e}_i]_l, u_{\mathrm{y}_j\mathrm{s}_j}\left(\sum_{k\neq l}[\hat{\boldsymbol{\beta}}_{-j}]_k[\mathbf{z}_j]_k + \hat{b}\right)[\mathbf{e}_j]_l\right)$$

$$=\mathrm{Cov}\left(u_{\mathrm{y}_i\mathrm{s}_i}\left(\sum_{k\neq l}[\hat{\boldsymbol{\beta}}_{-i}^{[jl]}]_k[\mathbf{z}_i]_k + \hat{b}\right)[\mathbf{e}_i]_l, u_{\mathrm{y}_j\mathrm{s}_j}\left(\sum_{k\neq l}[\hat{\boldsymbol{\beta}}_{-j}^{[il]}]_k[\mathbf{z}_j]_k + \hat{b}\right)[\mathbf{e}_j]_l\right)$$

$$+\mathrm{Cov}\left(u'_{\mathrm{y}_i\mathrm{s}_i}\left(\sum_{k\neq l}[\hat{\boldsymbol{\beta}}_{-i}^{[jl]}]_k[\mathbf{z}_i]_k + \hat{b}\right)\frac{r}{n}[\mathbf{e}_i]_l[\mathbf{e}_j]_l, u_{\mathrm{y}_j\mathrm{s}_j}\left(\sum_{k\neq l}[\hat{\boldsymbol{\beta}}_{-j}^{[il]}]_k[\mathbf{z}_j]_k + \hat{b}\right)[\mathbf{e}_j]_l\right)$$

$$+\mathrm{Cov}\left(u_{\mathrm{y}_i\mathrm{s}_i}\left(\sum_{k\neq l}[\hat{\boldsymbol{\beta}}_{-i}^{[jl]}]_k[\mathbf{z}_i]_k + \hat{b}\right)[\mathbf{e}_i]_l, u'_{\mathrm{y}_j\mathrm{s}_j}\left(\sum_{k\neq l}[\hat{\boldsymbol{\beta}}_{-j}^{[il]}]_k[\mathbf{z}_j]_k + \hat{b}\right)[\mathbf{e}_j]_l\frac{r}{n}[\mathbf{e}_j]_l[\mathbf{e}_i]_l\right) + o(n^{-1})$$

$$=o(n^{-1}).$$

Therefore,

$$\mathrm{Var}(\mathbf{w}_l) = \frac{1}{n^2}\sum_{i=1}^{n}\mathrm{Var}\left(u_{\mathrm{y}_i\mathrm{s}_i}\left(\sum_{k\neq l}[\hat{\boldsymbol{\beta}}_{-i}]_k[\mathbf{z}_i]_k + \hat{b}\right)[\mathbf{e}_i]_l\right) + o(n^{-1})$$

$$= \frac{\sigma_l}{n^2}\|\hat{\boldsymbol{\alpha}}\|^2 + o(n^{-1}).$$

We thus obtain that

$$\mathrm{Var}(\mathbf{z}^\mathsf{T}\mathbf{Q}(\nu)\mathbf{w}) = \frac{1}{n^2}\|\hat{\boldsymbol{\alpha}}\|^2\|\mathbf{Q}(\nu)\mathbf{\Sigma}\|_2^2 + o(1),$$

which concludes the proof of Eq. (9).

It remains to determine the value of $\kappa$. To see this, let us define $\mathbf{w}_{-i}$ with

$$[\mathbf{w}_{-i}]_l = \frac{1}{n}\sum_{j\neq i} u_{\mathrm{y}_j \mathrm{s}_j}\bigg(\sum_{k\neq l}[\hat{\boldsymbol{\beta}}_{-j}]_k[\mathbf{z}_j]_k + \hat{b}\bigg)[\mathbf{e}_j]_l.$$

Note that

$$\mathbf{z}_i^\mathsf{T}\mathbf{Q}(\nu)(\mathbf{w}-\mathbf{w}_{-i}) = \frac{1}{n}\sum_{l=1}^{p} u_{\mathrm{y}_i \mathrm{s}_i}\bigg(\sum_{k\neq l}[\hat{\boldsymbol{\beta}}_{-i}]_k[\mathbf{z}_i]_k + \hat{b}\bigg)[\mathbf{e}_i]_l[\mathbf{Q}(\nu)]_{ll}[\mathbf{z}_i]_l = \frac{\hat{\alpha}_i}{n}\,\mathrm{Tr}\,\mathbf{Q}(\nu)\mathbf{\Sigma} + o(1).$$

Using manipulations with $\hat{\boldsymbol{\beta}}_{-j}^{[il]}$ as before, we have

$$\mathrm{Var}(\mathbf{z}_i^\mathsf{T}\mathbf{Q}(\nu)\mathbf{w}_{-i}) = \frac{1}{n^2}\|\hat{\boldsymbol{\alpha}}\|^2\|\mathbf{Q}(\nu)\mathbf{\Sigma}\|_2^2 + o(1).$$

Finally, we get from Eq. (37) that

$$\mathrm{Var}\bigg(\hat{\boldsymbol{\beta}}^\mathsf{T}\mathbf{z}_i - \frac{\hat{\alpha}_i}{n}\,\mathrm{Tr}\,\mathbf{Q}(\nu)\mathbf{\Sigma}\bigg|\mathrm{y}_i,\mathrm{s}_i\bigg) = \mathrm{Var}(\hat{\boldsymbol{\beta}}^\mathsf{T}\mathbf{z}|\mathrm{y},\mathrm{s}) + o(1) = \mathrm{Var}(\hat{\boldsymbol{\beta}}_{-i}^\mathsf{T}\mathbf{z}_i|\mathrm{y}_i,\mathrm{s}_i) + o(1).$$

Comparing the above equation with Eq. (28), we find that

$$\kappa = \frac{1}{n}\,\mathrm{Tr}\,\mathbf{Q}(\nu)\mathbf{\Sigma} + o(1).$$

Remark also from Eq. (27) that $\nu$ is positive, and since $\kappa = \Theta(1)$ in the considered overparameterized regime with $p > n$, the value of $\nu$ is bounded. We conclude the proof of Theorem 4.4 by setting $\kappa = \frac{1}{n}\,\mathrm{Tr}\,\mathbf{Q}(\nu)\mathbf{\Sigma}$.

### A.4. Proof of Corollary 5.1

From Eq. (11), we get $\eta_{\mathrm{y}1} - \eta_{\mathrm{y}0} \xrightarrow{P} 0$ as $\Delta\widehat{\mathbb{E}}[\hat{\alpha}|\mathrm{s}] \xrightarrow{P} 0$. Then, according to Eq. (9), we have

$$\mathbb{E}[h(f_{\hat{\boldsymbol{\theta}}}(\mathbf{x}))|\mathrm{y},\mathrm{s}] - \mathbb{E}[h(f_{\hat{\boldsymbol{\theta}}}(\mathbf{x}))|\mathrm{y}] \to 0,$$

for all bounded Lipschitz $h\colon \mathbb{R} \mapsto \mathbb{R}$. In other words, $f_{\hat{\boldsymbol{\theta}}}(\mathbf{x}))$ is asymptotically independent of s when conditioned on y. The property of strong equalized odds defined in Eq. (2) is therefore achieved asymptotically.

Letting $h(r) = r$ in Eq. (10), we obtain from Eq. (11) that

$$\mathbb{E}[f_{\hat{\boldsymbol{\theta}}}(\mathbf{x})|\mathrm{y},\mathrm{s}=1] - \mathbb{E}[f_{\hat{\boldsymbol{\theta}}}(\mathbf{x})|\mathrm{y},\mathrm{s}=0] = \eta_{\mathrm{y}0} - \eta_{\mathrm{y}1} + o(1) = \boldsymbol{\mu}_{\mathcal{S}^c}^\mathsf{T}\tilde{\boldsymbol{\beta}}_{\mathcal{S}^c}\Delta\widehat{\mathbb{E}}[\hat{\alpha}|\mathrm{s}] + o(1),$$

where we remark that $\boldsymbol{\mu}_{\mathcal{S}^c}^\mathsf{T}\tilde{\boldsymbol{\beta}}_{\mathcal{S}^c} = \Theta(1)$ as a consequence of Assumption 4.2 and the boundedness of $\nu$. We therefore note that $\Delta\widehat{\mathbb{E}}[\hat{\alpha}|\mathrm{s}] \xrightarrow{P} 0$ is a necessary condition for the conditional independence between $f_{\hat{\boldsymbol{\theta}}}(\mathbf{x})$ and s to hold asymptotically.

### A.5. Proof of Theorem 5.2

Taking $h(r) = r$ in Eq. (10) and Eq. (9), we get

$$\widehat{\mathbb{E}}[f_{\hat{\boldsymbol{\theta}}}(\mathbf{x})|\mathrm{y},\mathrm{s}] - \kappa\widehat{\mathbb{E}}[\hat{\alpha}|\mathrm{y},\mathrm{s}] - \eta_{\mathrm{ys}} \xrightarrow{P} 0,$$

which leads to

$$\Delta\widehat{\mathbb{E}}[f_{\hat{\boldsymbol{\theta}}}(\mathbf{x})|\mathrm{y}=y,\mathrm{s}] - \kappa\Delta\widehat{\mathbb{E}}[\hat{\alpha}|\mathrm{y}=y,\mathrm{s}] - (\eta_{\mathrm{y}1} - \eta_{\mathrm{y}0}) \xrightarrow{P} 0, \quad \forall y \in \{0,1\}.$$

When $\Delta\widehat{\mathbb{E}}[\hat{\alpha}|s] \overset{P}{\to} 0$, we have $\eta_{y1} - \eta_{y0} \overset{P}{\to} 0$ according to Eq. (11). Then the constraint in Eq. (4), which imposes

$$\Delta\widehat{\mathbb{E}}[f_{\hat{\theta}}(\mathbf{x})|\mathbf{y} = y, s] + \omega_y\Delta\widehat{\mathbb{E}}[f_{\theta}(\mathbf{x})|\mathbf{y} = y] = 0,$$

results in

$$\kappa\Delta\widehat{\mathbb{E}}[\hat{\alpha}|\mathbf{y} = y, s] + \omega_y\Delta\widehat{\mathbb{E}}[f_{\theta}(\mathbf{x})|\mathbf{y} = y] \overset{P}{\to} 0.$$

Therefore, if $\Delta\widehat{\mathbb{E}}[\hat{\alpha}|s] \overset{P}{\to} 0$, we have also

$$\kappa\Delta\widehat{\mathbb{E}}[\hat{\alpha}|\mathbf{y} = y, s] + \omega_y\Delta\widehat{\mathbb{E}}[f_{\theta}(\mathbf{x})|\mathbf{y}] \overset{P}{\to} 0, \tag{39}$$

thereby demonstrating Eq. (12).

Now we will prove Eq. (13). Note first that

$$\widehat{\mathbb{E}}[\hat{\alpha}|s = 1] - \widehat{\mathbb{E}}[\hat{\alpha}|s = 0] = \widehat{P}(\mathbf{y} = 0|s = 1)\widehat{\mathbb{E}}[\hat{\alpha}|\mathbf{y} = 0, s = 1] + \widehat{P}(\mathbf{y} = 1|s = 1)\widehat{\mathbb{E}}[\hat{\alpha}|\mathbf{y} = 1, s = 1]$$
$$- \widehat{P}(\mathbf{y} = 0|s = 0)\widehat{\mathbb{E}}[\hat{\alpha}|\mathbf{y} = 0, s = 0] - \widehat{P}(\mathbf{y} = 1|s = 0)\widehat{\mathbb{E}}[\hat{\alpha}|\mathbf{y} = 1, s = 0],$$

and

$$\widehat{\mathbb{E}}[\hat{\alpha}|\mathbf{y} = 1] - \widehat{\mathbb{E}}[\hat{\alpha}|\mathbf{y} = 0] = \widehat{P}(s = 0|\mathbf{y} = 1)\widehat{\mathbb{E}}[\hat{\alpha}|\mathbf{y} = 1, s = 0] + \widehat{P}(s = 1|\mathbf{y} = 1)\widehat{\mathbb{E}}[\hat{\alpha}|\mathbf{y} = 1, s = 1]$$
$$- \widehat{P}(s = 0|\mathbf{y} = 0)\widehat{\mathbb{E}}[\hat{\alpha}|\mathbf{y} = 0, s = 0] - \widehat{P}(s = 1|\mathbf{y} = 0)\widehat{\mathbb{E}}[\hat{\alpha}|\mathbf{y} = 0, s = 1].$$

It follows that

$$\widehat{\mathbb{E}}[\hat{\alpha}|s = 1] - \widehat{\mathbb{E}}[\hat{\alpha}|s = 0] - \gamma\left(\widehat{\mathbb{E}}[\hat{\alpha}|\mathbf{y} = 1] - \widehat{\mathbb{E}}[\hat{\alpha}|\mathbf{y} = 0]\right) = \left[\widehat{P}(\mathbf{y} = 0|s = 1) + \gamma\widehat{P}(s = 1|\mathbf{y} = 0)\right]\widehat{\mathbb{E}}[\hat{\alpha}|\mathbf{y} = 0, s = 1]$$
$$+ \left[\widehat{P}(\mathbf{y} = 1|s = 1) - \gamma\widehat{P}(s = 1|\mathbf{y} = 1)\right]\widehat{\mathbb{E}}[\hat{\alpha}|\mathbf{y} = 1, s = 1] - \left[\widehat{P}(\mathbf{y} = 0|s = 0) - \gamma\widehat{P}(s = 0|\mathbf{y} = 0)\right]\widehat{\mathbb{E}}[\hat{\alpha}|\mathbf{y} = 0, s = 0]$$
$$- \left[\widehat{P}(\mathbf{y} = 1|s = 0) + \gamma\widehat{P}(s = 0|\mathbf{y} = 1)\right]\widehat{\mathbb{E}}[\hat{\alpha}|\mathbf{y} = 1, s = 0], \tag{40}$$

where we recall from Eq. (8) that $\gamma := \widehat{P}(\mathbf{y} = 1|s = 1) - \widehat{P}(\mathbf{y} = 1|s = 0)$. Remark also that

$$\widehat{P}(\mathbf{y} = 1|s = 1) - \gamma\widehat{P}(s = 1|\mathbf{y} = 1)$$
$$= \widehat{P}(\mathbf{y} = 1|s = 1) - \widehat{P}(\mathbf{y} = 1|s = 1)\widehat{P}(s = 1|\mathbf{y} = 1) + \widehat{P}(\mathbf{y} = 1|s = 0)\widehat{P}(s = 1|\mathbf{y} = 1)$$
$$= \widehat{P}(\mathbf{y} = 1|s = 1)\widehat{P}(s = 0|\mathbf{y} = 1) + \widehat{P}(\mathbf{y} = 1|s = 0)\widehat{P}(s = 1|\mathbf{y} = 1),$$

and

$$\widehat{P}(\mathbf{y} = 1|s = 0) + \gamma\widehat{P}(s = 0|\mathbf{y} = 1)$$
$$= \widehat{P}(\mathbf{y} = 1|s = 0) + \widehat{P}(\mathbf{y} = 1|s = 1)\widehat{P}(s = 0|\mathbf{y} = 1) - \widehat{P}(\mathbf{y} = 1|s = 0)\widehat{P}(s = 0|\mathbf{y} = 1)$$
$$= \widehat{P}(\mathbf{y} = 1|s = 1)\widehat{P}(s = 0|\mathbf{y} = 1) + \widehat{P}(\mathbf{y} = 1|s = 0)\widehat{P}(s = 1|\mathbf{y} = 1),$$

which leads to

$$\widehat{P}(\mathbf{y} = 1|s = 1) - \gamma\widehat{P}(s = 1|\mathbf{y} = 1) = \widehat{P}(\mathbf{y} = 1|s = 0) + \gamma\widehat{P}(s = 0|\mathbf{y} = 1) = \zeta_1, \tag{41}$$

with $\zeta_1 := \frac{\prod_{s=0}^{1}\widehat{P}(\mathbf{y}=1,s=s)}{\widehat{P}(\mathbf{y}=1)}\sum_{s=0}^{1}\frac{1}{\widehat{P}(s=s)}$. Similarly, we have

$$\widehat{P}(\mathbf{y} = 0|s = 1) + \gamma\widehat{P}(s = 1|\mathbf{y} = 0) = \widehat{P}(\mathbf{y} = 0|s = 0) - \gamma\widehat{P}(s = 0|\mathbf{y} = 0) = \zeta_0, \tag{42}$$

with $\zeta_0 := \frac{\prod_{s=0}^{1}\widehat{P}(\mathbf{y}=0,s=s)}{\widehat{P}(\mathbf{y}=0)}\sum_{s=0}^{1}\frac{1}{\widehat{P}(s=s)}$.

We prove Eq. (13) by plugging Eq. (41) and Eq. (42) into Eq. (40).

Now we are ready to prove the first part of Theorem 5.2 on the failure of fair ERM. It is easy to see from Eq. (9) and Eq. (11) that having $\mathbb{E}\left[h(f_{\hat{\boldsymbol{\theta}}}(\mathbf{x}))|\mathbf{y}, \mathbf{s}\right] - \mathbb{E}\left[h(f_{\hat{\boldsymbol{\theta}}}(\mathbf{x}))|\mathbf{y}\right] \to 0$ requires $\Delta\widehat{\mathbb{E}}[\hat{\alpha}|\mathbf{s}] \xrightarrow{P} 0$. Letting $\omega_0, \omega_1 = 0$ in Eq. (39), we get

$$\Delta\widehat{\mathbb{E}}[\hat{\alpha}|\mathbf{y} = y, \mathbf{s}] \xrightarrow{P} 0.$$

In other words, if we have $\Delta\widehat{\mathbb{E}}[\hat{\alpha}|\mathbf{s}] \xrightarrow{P} 0$ with $\omega_0, \omega_1 = 0$, then we also have $\Delta\widehat{\mathbb{E}}[\hat{\alpha}|\mathbf{y} = y, \mathbf{s}] \xrightarrow{P} 0$.

When $\Delta\widehat{\mathbb{E}}[\hat{\alpha}|\mathbf{s}] \xrightarrow{P} 0$ and $\Delta\widehat{\mathbb{E}}[\hat{\alpha}|\mathbf{y} = y, \mathbf{s}] \xrightarrow{P} 0$, Eq. (13) entails

$$\Delta\widehat{\mathbb{E}}[\hat{\alpha}|\mathbf{y}] \xrightarrow{P} 0,$$

which results in

$$\mathbb{E}\left[h(f_{\hat{\boldsymbol{\theta}}}(\mathbf{x}))|\mathbf{y}\right] - \mathbb{E}\left[h(f_{\hat{\boldsymbol{\theta}}}(\mathbf{x}))\right] \to 0,$$

as a consequence of Eq. (9) and Eq. (11).

It remains to prove the necessity of bias overcompensation as declared in the second part of Theorem 5.2. As mentioned above, having $\mathbb{E}\left[h(f_{\hat{\boldsymbol{\theta}}}(\mathbf{x}))|\mathbf{y}, \mathbf{s}\right] - \mathbb{E}\left[h(f_{\hat{\boldsymbol{\theta}}}(\mathbf{x}))|\mathbf{y}\right] \to 0$ requires $\Delta\widehat{\mathbb{E}}[\hat{\alpha}|\mathbf{s}] \xrightarrow{P} 0$. When $\Delta\widehat{\mathbb{E}}[\hat{\alpha}|\mathbf{s}] \xrightarrow{P} 0$, it can be seen from Eq. (9) and Eq. (11) that having $\liminf_{n \to \infty} \text{Cor}(f_{\hat{\boldsymbol{\theta}}}(\mathbf{x}), \mathbf{y}) > 0$ imposes $\Delta\widehat{\mathbb{E}}[\hat{\alpha}|\mathbf{y}]$ to be positive and bounded away from zero with high probability.

When $\Delta\widehat{\mathbb{E}}[\hat{\alpha}|\mathbf{s}] \xrightarrow{P} 0$, it follows from Eq. (13) that

$$\gamma\Delta\widehat{\mathbb{E}}[\hat{\alpha}|\mathbf{y}] + \sum_{y \in \{0,1\}} \zeta_y \Delta\mathbb{E}[\hat{\alpha}|\mathbf{y} = y, \mathbf{s}] \xrightarrow{P} 0.$$

Plugging Eq. (39) into the above convergence, we get

$$\gamma\Delta\widehat{\mathbb{E}}[\hat{\alpha}|\mathbf{y}] - \frac{\zeta_0\omega_0 + \zeta_1\omega_1}{\kappa}\Delta\widehat{\mathbb{E}}[f_{\boldsymbol{\theta}}(\mathbf{x})|\mathbf{y}] \xrightarrow{P} 0.$$

From Eq. (10) and Eq. (11), we observe that, when $\Delta\widehat{\mathbb{E}}[\hat{\alpha}|\mathbf{s}] \xrightarrow{P} 0$,

$$\Delta\widehat{\mathbb{E}}[f_{\boldsymbol{\theta}}(\mathbf{x})|\mathbf{y}] - (\eta + \xi)\Delta\widehat{\mathbb{E}}[\hat{\alpha}|\mathbf{y}] \xrightarrow{P} 0.$$

where $\xi = \boldsymbol{\mu}_{\mathcal{S}}^{\mathsf{T}}\tilde{\boldsymbol{\beta}}_{\mathcal{S}}$ Combining the last two convergences in separate lines, we obtain that

$$\zeta_0\omega_0 + \zeta_1\omega_1 - \frac{\gamma\kappa}{\xi + \kappa} \xrightarrow{P} 0, \tag{43}$$

which implies $\zeta_0\omega_0 + \zeta_1\omega_1 > 0$, since we have $\gamma > 0$ from Eq. (8), and $\kappa, \xi > 0$ form Theorem 4.4. As $\zeta_0, \zeta_1 > 0$, we conclude that at least one of $\omega_0, \omega_1$ is strictly positive.

### A.6. Proof of Theorem 5.3

Let us set $\omega_{\mathbf{y}} = \rho_{\mathbf{y}}\omega$ with

$$\rho_{\mathbf{y}} = \frac{(2\mathbf{y} - 1)\left(\frac{1}{n_{\mathbf{y}0}} - \frac{1}{n_{\mathbf{y}1}}\right)}{\frac{1}{n_{10}} - \frac{1}{n_{11}} - \frac{1}{n_{00}} + \frac{1}{n_{01}}}.$$

Then from Eq. (43), we have

$$(\zeta_0\rho_0 + \zeta_1\rho_1)\omega - \frac{\gamma\kappa}{\xi + \kappa} \xrightarrow{P} 0.$$

Since $\zeta_0, \zeta_1, \rho_0, \rho_1, \gamma$ can be calculated directly from the cardinalities $n_{00}, n_{01}, n_{10}, n_{11}$, we will focus on $\frac{\kappa}{\xi + \kappa}$.

Recall that

$$\kappa = \frac{1}{n}\text{Tr}\,\mathbf{Q}(\nu)\boldsymbol{\Sigma}, \quad \xi = \boldsymbol{\mu}_{\mathcal{S}}^{\mathsf{T}}\tilde{\boldsymbol{\beta}}_{\mathcal{S}} = \prod_{y=0}^{1}\widehat{P}(\mathbf{y} = y)\boldsymbol{\mu}_{\mathcal{S}}^{\mathsf{T}}[\mathbf{Q}(\nu)]_{\mathcal{S}\mathcal{S}}\boldsymbol{\mu}_{\mathcal{S}}.$$

Since $\frac{\partial \kappa}{\partial \nu}, \frac{\partial \xi}{\partial \nu}, \frac{\partial \frac{\kappa}{\xi}}{\partial \nu} < 0$, we remark that $\kappa, \xi, \frac{\kappa}{\xi}$ are decreasing functions of $\nu$. Therefore, we have

$$\kappa \leq \kappa_{\sup} := \frac{1}{n} \operatorname{Tr} \mathbf{Q}(0) \mathbf{\Sigma} = \frac{1}{n} \operatorname{Tr} \mathbf{\Sigma},$$

$$\xi \leq \xi_{\sup} = \prod_{y=0}^{1} \widehat{P}(\mathbf{y} = y) \boldsymbol{\mu}_{\mathcal{S}}^{\mathsf{T}} [\mathbf{Q}(0)]_{\mathcal{S}\mathcal{S}} \boldsymbol{\mu}_{\mathcal{S}} = \prod_{y=0}^{1} \widehat{P}(\mathbf{y} = y) \|\boldsymbol{\mu}_{\mathcal{S}}\|^2,$$

and

$$\frac{\kappa}{\xi} \leq \frac{\kappa_{\sup}}{\xi_{\sup}}.$$

To estimate $\kappa_{\sup}, \xi_{\sup}$ from the training data, we propose the following estimators:

$$\hat{\kappa}_{\sup} = \frac{1}{n^2} \operatorname{Tr} \mathbf{K}, \quad \hat{\eta}_{\sup} = \frac{(n_{10} + n_{11})(n_{00} + n_{01})}{n^2} \left( \mathbf{v}^{\mathsf{T}} \mathbf{K} \mathbf{v} - \hat{\kappa}_{\sup} \sum_{y,s \in \{0,1\}} \frac{n}{4n_{ys}} \right), \tag{44}$$

where $\mathbf{v} = \frac{1}{2n_{11}} \mathbf{q}_{11} + \frac{1}{2n_{10}} \mathbf{q}_{10} - \frac{1}{2n_{01}} \mathbf{q}_{01} - \frac{1}{2n_{00}} \mathbf{q}_{00}$. Since $\frac{1}{n} \|\mathbf{z}_i\|^2 = \frac{1}{n} \operatorname{Tr} \mathbf{\Sigma} + O(n^{-\frac{1}{2}})$, we have

$$\hat{\kappa}_{\sup} = \frac{1}{n^2} \operatorname{Tr} \mathbf{K} = \frac{1}{n^2} \sum_{i=1}^{n} \|\mathbf{z}_i\|^2 = \frac{1}{n} \operatorname{Tr} \mathbf{\Sigma} + O(n^{-1}).$$

Note also that

$$\mathbf{Z}\mathbf{v} = \begin{bmatrix} \boldsymbol{\mu}_{\mathcal{S}} \\ \mathbf{0}_{|\mathcal{S}^c|} \end{bmatrix} + \mathbf{E}\mathbf{v},$$

where $\mathbf{E} = [\mathbf{e}_1, \ldots, \mathbf{e}_n]$. Then we obtain that

$$\mathbf{v}^{\mathsf{T}} \mathbf{K} \mathbf{v} = \|\boldsymbol{\mu}_{\mathcal{S}}\|^2 + \mathbf{v}^{\mathsf{T}} \mathbf{E}^{\mathsf{T}} \mathbf{E} \mathbf{v} + O(n^{-\frac{1}{2}}) = \|\boldsymbol{\mu}_{\mathcal{S}}\|^2 + \|\mathbf{v}\|^2 \operatorname{Tr} \mathbf{K} + O(n^{-\frac{1}{2}}) = \|\boldsymbol{\mu}_{\mathcal{S}}\|^2 + \hat{\kappa}_{\sup} \sum_{y,s \in \{0,1\}} \frac{n}{4n_{ys}} + O(n^{-\frac{1}{2}}).$$

Now we turn to the infima. Remark from Eq. (27) that

$$\kappa \geq \lim_{a \to +\infty} \frac{1}{n} \operatorname{Tr} \mathbf{\Sigma} \left( \mathbf{I}_p + \frac{s}{n} \mathbf{Z}\mathbf{Z}^{\mathsf{T}} \right)^{-1} = \frac{1}{n} \operatorname{Tr} \mathbf{\Sigma} \left( \mathbf{I}_p - \mathbf{Z} \left( \mathbf{Z}^{\mathsf{T}} \mathbf{Z} \right)^{-1} \mathbf{Z}^{\mathsf{T}} \right) := \kappa_{\inf}.$$

Note also that

$$n[\mathbf{K}^{-1}]_{ii} = n \left( \mathbf{z}_i^{\mathsf{T}} \left( \mathbf{I}_p - \mathbf{Z}_{-i} \left( \mathbf{Z}_{-i}^{\mathsf{T}} \mathbf{Z}_{-i} \right)^{-1} \mathbf{Z}_{-i}^{\mathsf{T}} \right) \mathbf{z}_i \right)^{-1}$$

where $\mathbf{Z}_{-i} = [\mathbf{z}_1, \ldots, \mathbf{z}_{i-1}, \mathbf{z}_{i+1}, \ldots, \mathbf{z}_n]$. We then set

$$\hat{\kappa}_{\inf} = \frac{1}{n} \sum_{i=1}^{n} \frac{1}{n[\mathbf{K}^{-1}]_{ii}},$$

for which we have $\hat{\kappa}_{\inf} = \kappa_{\inf} + O(n^{-1})$.

Recall from Eq. (27) that $\kappa = \frac{1}{n} \operatorname{Tr} \mathbf{\Sigma} \left( \mathbf{I}_p + \frac{1}{\lambda n} \mathbf{Z} \operatorname{diag}(\hat{\mathbf{t}}) \mathbf{Z}^{\mathsf{T}} \right)^{-1}$ with $\hat{\mathbf{t}} := \{\ell'' \left( \frac{1}{n}[\mathbf{K}\hat{\boldsymbol{\alpha}}]_i + \hat{b}, \mathbf{y}_i \right)\}_{i=1}^{n}$. We observe that when square loss $\ell(a, b) = (a - b)^2$ is used, $\hat{\mathbf{t}} = 2\mathbf{1}_n$, then

$$\lim_{\lambda \to 0} \kappa = \lim_{\lambda \to 0} \frac{1}{n} \operatorname{Tr} \mathbf{\Sigma} \left( \mathbf{I}_p + \frac{2}{\lambda n} \mathbf{Z}\mathbf{Z}^{\mathsf{T}} \right)^{-1} = \kappa_{\inf}.$$

Therefore, we have $\kappa = \kappa_{\inf}$, and correspondingly $\xi = \xi_{\inf}$, when $\xi, \kappa$ are given by the unregularized least-square solution $\hat{\boldsymbol{\alpha}}_{\mathrm{LS}} = n\mathbf{K}^{-1}\mathbf{y}$ where $\mathbf{y} = \{\mathbf{y}_i\}_{i=1}^{n}$. Applying Eq. (9), we obtain

$$\xi = \frac{\mathbb{E}\left[ f_{\hat{\boldsymbol{\theta}}}(\mathbf{x}) \middle| \mathbf{y} = 1, \mathbf{s} \right] - \mathbb{E}\left[ f_{\hat{\boldsymbol{\theta}}}(\mathbf{x}) \middle| \mathbf{y} = 0, \mathbf{s} \right]}{\Delta \widehat{\mathbb{E}}[\hat{\alpha} | \mathbf{y}]} + o(1).$$

Then, using Eq. (10), we get

$$\xi = \frac{\widehat{\mathbb{E}}\big[f_{\hat{\boldsymbol{\theta}}}(\mathbf{x})\big|\mathbf{y}=1,\mathbf{s}\big] - \widehat{\mathbb{E}}\big[f_{\hat{\boldsymbol{\theta}}}(\mathbf{x})\big|\mathbf{y}=0,\mathbf{s}\big] - \kappa\left(\widehat{\mathbb{E}}[\hat{\alpha}|\mathbf{y}=1,\mathbf{s}] - \widehat{\mathbb{E}}[\hat{\alpha}|\mathbf{y}=0,\mathbf{s}]\right)}{\Delta\widehat{\mathbb{E}}[\hat{\alpha}|\mathbf{y}]} + o(1).$$

Notice also that for $\hat{\boldsymbol{\alpha}}_{\mathrm{LS}} = n\mathbf{K}^{-1}\mathbf{y}$, we have $\widehat{\mathbb{E}}\big[f_{\hat{\boldsymbol{\theta}}}(\mathbf{x})\big|\mathbf{y}=1,\mathbf{s}\big] - \widehat{\mathbb{E}}\big[f_{\hat{\boldsymbol{\theta}}}(\mathbf{x})\big|\mathbf{y}=0,\mathbf{s}\big] = 1$. Therefore,

$$\xi_{\mathrm{inf}} = \frac{1 - \kappa_{\mathrm{inf}}\left(\widehat{\mathbb{E}}\hat{\alpha}_{\mathrm{LS}}|\mathbf{y}=1,\mathbf{s}] - \widehat{\mathbb{E}}[\hat{\alpha}_{\mathrm{LS}}|\mathbf{y}=0,\mathbf{s}]\right)}{\Delta\widehat{\mathbb{E}}[\hat{\alpha}_{\mathrm{LS}}|\mathbf{y}]} + o(1).$$

Setting

$$\hat{\xi}_{\mathrm{inf}} = \left(1 - \frac{\Delta\widehat{\mathbb{E}}[\hat{\alpha}_{\mathrm{LS}}|\mathbf{y},\mathbf{s}=0] + \Delta\widehat{\mathbb{E}}[\hat{\alpha}_{\mathrm{LS}}|\mathbf{y},\mathbf{s}=1]}{2}\kappa_{\mathrm{inf}}\right)\frac{1}{\Delta\widehat{\mathbb{E}}[\hat{\alpha}_{\mathrm{LS}}|\mathbf{y}]},$$

we obtain $\hat{\xi}_{\mathrm{inf}} = \xi_{\mathrm{inf}} + o(1)$. Finally, if a non-trivial fair classification is achieved in the sense of Eq. (14), we have

$$\omega \in [\hat{\omega}_{\mathrm{inf}}, \hat{\omega}_{\mathrm{sup}}]$$

with high probability, where $\hat{\omega}_{\mathrm{sup}} = \frac{\gamma\hat{\kappa}_{\mathrm{sup}}}{\hat{\kappa}_{\mathrm{sup}}+\hat{\xi}_{\mathrm{sup}}}\frac{1}{\zeta_0\rho_0+\zeta_1\rho_1}$ and $\hat{\omega}_{\mathrm{inf}} = \frac{\gamma\hat{\kappa}_{\mathrm{inf}}}{\hat{\kappa}_{\mathrm{inf}}+\hat{\xi}_{\mathrm{inf}}}\frac{1}{\zeta_0\rho_0+\zeta_1\rho_1}$.

### A.7. Proof of Theorem 6.1

Similarly to the proof in Appendix A.3, we start by defining the Lagrangian function

$$L(\boldsymbol{\alpha}, b, \psi) := q(\boldsymbol{\alpha}, b) + \frac{\psi}{n}\mathbf{u}^{\mathsf{T}}\boldsymbol{\alpha}$$

where $q(\boldsymbol{\alpha}, b) := \frac{1}{n}\sum_{i=1}^{n}\ell\left(\frac{1}{n}[\mathbf{K}\boldsymbol{\alpha}]_i + b, \mathbf{y}_i\right) + \frac{\lambda}{2n^2}\boldsymbol{\alpha}^{\mathsf{T}}\mathbf{K}\boldsymbol{\alpha}$ is the objective function, and $\psi$ are the Lagrange multipliers for the linear constraint on balanced coefficients in Eq. (6), which we rewrite as $\frac{1}{n}\mathbf{u}^{\mathsf{T}}\boldsymbol{\alpha} = 0$ with

$$\mathbf{u} = n\left(\frac{(\mathbf{q}_{01}+\mathbf{q}_{11})}{n_{01}+n_{11}} - \frac{(\mathbf{q}_{00}+\mathbf{q}_{10})}{n_{00}+n_{10}}\right).$$

Canceling the partial derivatives, we get

$$\frac{\partial L(\hat{\boldsymbol{\alpha}}, \hat{b}, \hat{\psi})}{\partial\hat{\boldsymbol{\alpha}}} = -\frac{1}{n^2}\mathbf{K}\hat{\mathbf{a}} + \frac{\lambda}{n^2}\mathbf{K}\hat{\boldsymbol{\alpha}} + \frac{\hat{\psi}}{n}\mathbf{u} = 0$$

$$\frac{\partial L(\hat{\boldsymbol{\alpha}}, \hat{b}, \hat{\psi}_0, \hat{\psi}_1)}{\partial\hat{b}} = \frac{1}{n}\mathbf{1}_n^{\mathsf{T}}\hat{\mathbf{a}} = 0$$

$$\frac{\partial L(\hat{\boldsymbol{\alpha}}, \hat{b}, \hat{\psi}_0, \hat{\psi}_1)}{\partial\hat{\psi}} = \frac{1}{n}\mathbf{u}^{\mathsf{T}}\hat{\boldsymbol{\alpha}} = 0$$

where $\hat{\mathbf{a}} := \left\{-\ell'\left(\frac{1}{n}[\mathbf{K}\hat{\boldsymbol{\alpha}}]_i + \hat{b}, \mathbf{y}_i\right)\right\}_{i=1}^{n}$ with $\ell'(r, y) = \frac{\partial\ell(r,y)}{\partial r}$. The first equality leads to

$$\hat{\boldsymbol{\alpha}} = \frac{1}{\lambda}(\hat{\mathbf{a}} + \hat{\boldsymbol{\alpha}}^{\mathrm{s}})$$

where $\hat{\boldsymbol{\alpha}}^{\mathrm{s}} = -n\mathbf{K}^{-1}\mathbf{u}$ Note that $\hat{\boldsymbol{\alpha}}^{\mathrm{s}} = -n\mathbf{K}^{-1}\mathbf{u}$ is the unregularized least-squares solution obtained by minimizing $\frac{1}{n}\sum_{i=1}^{n}(f_{\boldsymbol{\theta}}(\mathbf{x}_i) + [\mathbf{u}]_i)^2$. Remark also that $[\mathbf{u}]_i = (2\mathbf{s}_i - 1)\frac{n}{n_{0\mathbf{s}_i}+n_{1\mathbf{s}_i}}$, $\hat{\boldsymbol{\alpha}}^{\mathrm{s}}$ provides thus a prediction for $-\mathbf{s}$.

Similarly to Eq. (31) in Appendix A.3, it can be shown that

$$\hat{\alpha}_i = u(\hat{\boldsymbol{\beta}}_{-i}^{\mathsf{T}}\mathbf{z}_i + \hat{b} + \kappa\hat{\alpha}_i^{\mathrm{s}}) + O(n^{-\frac{1}{2}}).$$

with the mapping $u = \frac{1}{\kappa}\left(\text{prox}_{\ell(\cdot,y),\kappa/\lambda}(x) - x\right)$ for some $\kappa = \Theta(1)$ and $\text{prox}_{f,\tau}(x) = \arg\min_{t \in \mathbb{R}} f(t) + \frac{\tau}{2}\|t - x\|^2$. Analogously, for $\hat{\alpha}_i^s$, we have

$$\hat{\alpha}_i^s = u^s(\mathbf{z}_i^\mathsf{T}\hat{\boldsymbol{\beta}}_{-i}^s + \hat{b}^s) + O(n^{-\frac{1}{2}}).$$

Then following the reasoning leading to Eq. (37), we get

$$(\boldsymbol{I}_p + \nu\boldsymbol{\Sigma})\hat{\boldsymbol{\beta}} = \begin{bmatrix} \prod_{y=0}^{1}\widehat{P}(\mathrm{y}=y)\Delta\widehat{\mathbb{E}}[\hat{\alpha}|\mathrm{y}]\boldsymbol{\mu}_{\mathcal{S}} \\ \prod_{s=0}^{1}\widehat{P}(\mathrm{s}=s)\Delta\widehat{\mathbb{E}}[\hat{\alpha}|\mathrm{s}]\boldsymbol{\mu}_{\mathcal{S}^c} \end{bmatrix} + \nu'\boldsymbol{\Sigma}\hat{\boldsymbol{\beta}}^s + \mathbf{w} + o_{\|\cdot\|}(1),$$

for some constant $\nu, \nu' = \Theta(1)$, and random noise $\mathbf{w} = \Theta_{\|\cdot\|}(1)$. Since $\Delta\widehat{\mathbb{E}}[\hat{\alpha}|\mathrm{s}] = 0$ is imposed by Eq. (15), we have

$$\mathbb{E}[\mathbf{z}^\mathsf{T}\hat{\boldsymbol{\beta}}|\mathrm{y},\mathrm{s}] = \mathrm{y}\prod_{y=0}^{1}\widehat{P}(\mathrm{y}=y)\Delta\widehat{\mathbb{E}}[\hat{\alpha}|\mathrm{y}]\boldsymbol{\mu}_{\mathcal{S}}^\mathsf{T}\left[(\boldsymbol{I}_p + \nu\boldsymbol{\Sigma})^{-1}\right]_{\mathcal{S}\mathcal{S}}\boldsymbol{\mu}_{\mathcal{S}} + \mathrm{y}\boldsymbol{\mu}_{\mathcal{S}}^\mathsf{T}\left[(\boldsymbol{I}_p + \nu\boldsymbol{\Sigma})^{-1}\nu'\boldsymbol{\Sigma}\right]_{\mathcal{S}\mathcal{S}}\hat{\boldsymbol{\beta}}_{\mathcal{S}}^s$$
$$+ \mathrm{s}\boldsymbol{\mu}_{\mathcal{S}^c}^\mathsf{T}\left[(\boldsymbol{I}_p + \nu\boldsymbol{\Sigma})^{-1}\nu'\boldsymbol{\Sigma}\right]_{\mathcal{S}^c\mathcal{S}^c}\hat{\boldsymbol{\beta}}_{\mathcal{S}^c}^s.$$

Since $\hat{\boldsymbol{\beta}}^s = \frac{1}{n}\mathbf{Z}\hat{\boldsymbol{\alpha}}^s = -\frac{1}{n}\mathbf{Z}\mathbf{K}^{-1}\mathbf{u}$, we note that $\boldsymbol{\mu}_{\mathcal{S}^c}^\mathsf{T}\left[(\boldsymbol{I}_p + \nu\boldsymbol{\Sigma})^{-1}\nu'\boldsymbol{\Sigma}\right]_{\mathcal{S}^c\mathcal{S}^c}\hat{\boldsymbol{\beta}}_{\mathcal{S}^c}^s - c_n = o(1)$ for some $c_n = \Theta(1)$ and $c_n < 0$. Consequently, we have

$$\mathbb{E}[\mathbf{z}^\mathsf{T}\hat{\boldsymbol{\beta}}|\mathrm{y},\mathrm{s}] - \mathbb{E}[\mathbf{z}^\mathsf{T}\hat{\boldsymbol{\beta}}|\mathrm{y}] = \Theta(1),$$

and

$$\mathbb{E}[\mathbf{z}^\mathsf{T}\hat{\boldsymbol{\beta}}|\mathrm{y},\mathrm{s}=0] > \mathbb{E}[\mathbf{z}^\mathsf{T}\hat{\boldsymbol{\beta}}|\mathrm{y}],$$

which translates to a favorable bias towards the group $\mathrm{s} = 0$, as empirically observed in Figure 5.

# B. Algorithms for Fair ERM with Bias Overcompensation

---

**Algorithm 1** Fair ERM with Bias Overcompensation

---

1: **Input:** training data $\{(\mathbf{x}_i, \mathrm{y}_i, \mathrm{s}_i)\}_{i=1}^n$, kernel $k$, loss function $\ell$, hyperparameters $\lambda \in \mathbb{R}_{>0}$, $\upsilon \in \mathbb{R}_{\geq 0}$, $T \in \mathbb{Z}_{>0}$.

2: Set $\mathbf{q}_{\mathrm{ys}} = \{\mathbf{1}_{\mathrm{y}_i=\mathrm{y}} \mathbf{1}_{\mathrm{s}_i=\mathrm{s}}\}_{i=1}^n$ and $n_{\mathrm{ys}} = \|\mathbf{q}_{\mathrm{ys}}\|_1$.

3: Compute $\mathbf{K} = \{k(\mathbf{x}_i, \mathbf{x}_j)\}_{i,j=1}^n$.

4: Compute $\rho_0, \rho_1, \omega_{\inf}, \omega_{\sup}$ with Algorithm 2, and calculate $\omega_{\mathrm{mid}} = (\omega_{\inf} + \omega_{\sup})/2$.

5: Build grid $G_\omega$ with $T$ evenly spaced values in $[\max\{\omega_{\inf} - \upsilon\omega_{\mathrm{mid}}, 0\}, \max\{\omega_{\inf}, \omega_{\sup}\} + \upsilon\omega_{\mathrm{mid}}]$

6: **for** $t = 1$ **to** $T$ **do**

7:      Solve Eq. (6) with $\omega_y = \rho_y\omega$ for $y \in \{0, 1\}$, and store the solution as $\hat{\boldsymbol{\alpha}}^{(t)}, \hat{b}^{(t)}$.

8:      Compute and store $\Delta\widehat{\mathbb{E}}[\hat{\alpha}^{(t)}|\mathrm{s}] = \left(\frac{\mathbf{q}_{01}+\mathbf{q}_{11}}{n_{01}+n_{11}} - \frac{\mathbf{q}_{00}+\mathbf{q}_{10}}{n_{00}+n_{10}}\right)^{\mathsf{T}} \hat{\boldsymbol{\alpha}}^{(t)}$.

9: **end for**

10: Find $t^\star = \arg\min_t \left|\Delta\widehat{\mathbb{E}}[\hat{\alpha}^{(t)}|\mathrm{s}]\right|$.

11: **Return** $\hat{\boldsymbol{\alpha}}^{(t^\star)}, \hat{b}^{(t^\star)}$.

---

**Algorithm 2** Search range for $\omega_0, \omega_1$

---

1: **Input:** kernel matrix $\mathbf{K}$, indicator vectors $\{\mathbf{q}_{ys}\}_{y,s=0}^1$, cardinalities $\{n_{ys}\}_{y,s=0}^1$.

2: Compute $\rho_y = \frac{(2y-1)\left(\frac{1}{n_{y0}} - \frac{1}{n_{y1}}\right)}{\frac{1}{n_{10}} - \frac{1}{n_{11}} - \frac{1}{n_{00}} + \frac{1}{n_{01}}}$ for $y \in \{0, 1\}$.

3: Compute $\kappa_{\sup} = \frac{1}{n^2} \operatorname{Tr} \mathbf{K}$.

4: Set $\mathbf{g} = \frac{\mathbf{q}_{11}}{2n_{11}} + \frac{\mathbf{q}_{10}}{2n_{10}} - \frac{\mathbf{q}_{01}}{2n_{01}} - \frac{\mathbf{q}_{00}}{2n_{00}}$, and compute $\xi_{\sup} = \frac{(n_{00}+n_{01})(n_{10}+n_{11})}{n^2}\left(\mathbf{g}^{\mathsf{T}}\mathbf{K}\mathbf{g} - \kappa_{\sup}\sum_{y,s=0}^1 \frac{n}{4n_{ys}}\right)$.

5: Compute $\kappa_{\inf} = \frac{1}{n^2}\sum_{i=1}^n \frac{1}{[\mathbf{K}^{-1}]_{ii}}$.

6: Set $\hat{\boldsymbol{\alpha}} = n\mathbf{K}^{-1}(\mathbf{q}_{10} + \mathbf{q}_{11})$, and compute $\Delta\widehat{\mathbb{E}}[\hat{\alpha}|\mathrm{y}] = \left(\frac{\mathbf{q}_{10}+\mathbf{q}_{11}}{n_{10}+n_{11}} - \frac{\mathbf{q}_{00}+\mathbf{q}_{01}}{n_{00}+n_{01}}\right)^{\mathsf{T}}\hat{\alpha}$, $\Delta\widehat{\mathbb{E}}[\hat{\alpha}|\mathrm{y}, \mathrm{s} = s] = \left(\frac{\mathbf{q}_{1s}}{n_{1s}} - \frac{\mathbf{q}_{0s}}{n_{0s}}\right)^{\mathsf{T}}\hat{\alpha}$

     for $s \in \{0, 1\}$.

7: Compute $\xi_{\inf} = \left(1 - \frac{\Delta\widehat{\mathbb{E}}[\hat{\alpha}|\mathrm{y},\mathrm{s}=0]+\Delta\widehat{\mathbb{E}}[\hat{\alpha}|\mathrm{y},\mathrm{s}=1]}{2}\kappa_{\inf}\right)\frac{1}{\Delta\widehat{\mathbb{E}}[\hat{\alpha}|\mathrm{y}]}$

8: Set $\gamma = \frac{n_{11}}{n_{11}+n_{01}} - \frac{n_{10}}{n_{10}+n_{00}}$, and $\zeta_y = \frac{n_{y0}n_{y1}}{n_{y0}+n_{y1}}\left(\frac{1}{n_{00}+n_{10}} + \frac{1}{n_{01}+n_{11}}\right)$ for $y \in \{0, 1\}$,

9: Compute $\omega_{\sup} = \frac{\gamma\kappa_{\sup}}{\kappa_{\sup}+\xi_{\sup}}\frac{1}{\zeta_0\rho_0+\zeta_1\rho_1}$ and $\omega_{\inf} = \frac{\gamma\kappa_{\inf}}{\kappa_{\inf}+\xi_{\inf}}\frac{1}{\zeta_0\rho_0+\zeta_1\rho_1}$.

10: **Return** $\rho_0, \rho_1, \omega_{\inf}, \omega_{\sup}$.

---

