# OpenReview forum: "Correcting Overparameterization Effects in Fair Empirical Risk Minimization"
_ICML.cc/2026/Conference — ICML 2026 regular_

### Official Review · Reviewer_K975 · 2026-03-04

**Soundness:** 3
**Presentation:** 2
**Significance:** 3
**Originality:** 3
**Overall Recommendation:** 5
**Confidence:** 3

**Summary:**

**Summary:**  This paper addresses the problem of invoking fairness constraints in the ERM framework. IN particular, the authors claim that equalized odds (a notion of fairness) can be achieved in this framework by introducing bias overparameterization, which also turns out to be necessary in this case. Extensive simulation experiments presented in this paper support the theoretical claims made in this paper.

**Compliance With Llm Reviewing Policy:**

Affirmed.

**Key Questions For Authors:**

**Questions:**
 - According to Eqs (9) and (10), both $f_{\hat \theta}(x)$ and $f_{\hat \theta}(x)  - \kappa \hat \alpha_i$ have the same conditional distribution asymptotically. Why is there a single $i$? Also, I do not understand why $\kappa \hat \alpha_i$ is not affecting the distribution, as it is potentially a non-zero term. I might be missing something here.
 - Also, can you explain the importance of Thm 4.4 and how it play role in the overall contribution?

**Limitations:**

Yes

**Strengths And Weaknesses:**

**Strength:**
 - The paper addresses an important problem in fairness research and provides a practical solution to it.
 - The theoretical findings in this paper are interesting and hold value.

**Weakness:**
 - Could not find any apparent weaknesses. However, the authors missed a related reference [1] in the literature discussion in the introduction. This work theoretically explains the effect of overparameterization on mitigating bias in the random feature model under label shift.

[1] Roy, S., Maity, S., Xue, S., Yurochkin, M., & Sun, Y. (2025). How does overparametrization affect performance on minority groups?. Transactions on Machine Learning Research.

---

> ### Author Rebuttal · Authors · 2026-03-30
>
> We sincerely thank the reviewer for the time invested in our paper. Please find below our response.
>
> W1: Could not find any apparent weaknesses. However, the authors missed a related reference [1] in the literature discussion in the introduction. This work theoretically explains the effect of overparameterization on mitigating bias in the random feature model under label shift.
>
> We thank the reviewer for brining this closely related reference to our attention. We will include it in the revised version.
>
> Q1:  According to Eqs (9) and (10), ...
>
> We apologize for the confusion. There is a typo in Eq. (10), $f_{\hat{\theta}}(x)-\kappa\alpha_{i}$ should be changed to $f_{\hat{\theta}}(x)-\kappa\alpha$. With $\hat{\mathbb{E}}$ as understood in the first footnote at Page 2, Eq. (10) describes the empirical distribution of $\\{ f_{\hat{\theta}}(x_{i})-\kappa\alpha_{i}\| i\in[n] \\}$, which is asymptotically identical to the population distribution of $f_\hat\theta(x)$. In other words, the difference between training score $f_{\hat{\theta}}(x_i)$ and test score $f_{\hat{\theta}}(x)$ is captured by $\kappa\alpha_i$, which is a scaling of representer coefficients. In the revised version, we will correct this typo and move the footnote on the notation $\hat{\mathbb{E}}$ to the main text.
>
> Q2: Also, can you explain the importance of Thm 4.4 and how it play role in the overall contribution?
>
> Theorem 4.4 is the core of the paper, as it provides an exact characterization of the learning results (training and test scores) of fair ERM that is accessible via representer coefficients. The subsequent results of direct practical implications in Section 5 are derived based on that characterization. Precisely, Eq. (9) characterizes the test-score distribution in terms of the representer coefficients, and Eq. (10) explains how the training-score distribution differs from it through a self-influence term of the form $\kappa\alpha_i$. The offset $\hat{b}$ appears in $\eta_{ys}$ as a common additive shift, but it cancels out of fairness comparisons because it is not group-specific. This characterization through representer coefficients allow us to derive the balanced-coefficient criterion as a direct consequence of Eq. (9), the failure of fair ERM by demonstrating incompatible conditions on representer coefficients, and the need for bias overcompensation as an implication of the balanced-coefficient criterion on the empirical scores.

---

> > ### Author Rebuttal · Reviewer_K975 · 2026-04-01
> >
> > My concerns have been adequately addressed. I will keep my positive score.

---

> > > ### Author Response · Authors · 2026-04-05
> > >
> > > Thank you for acknowledging our reply and confirming your appreciation of our work.

---

### Official Review · Reviewer_AkbN · 2026-03-09

**Soundness:** 3
**Presentation:** 3
**Significance:** 2
**Originality:** 3
**Overall Recommendation:** 4
**Confidence:** 3

**Summary:**

This paper analyzes equalized odds (in a regression problem, as opposed to classification) under the setting where the learner defined on an RKHS with an offset (section 3.2), the data generating distribution is linear with a spurious correlation with the (binary) sensitive attribute $s$ (section 3.3), and the learning setting is overparameterized (i.e., p > n).

Under this specific setting:
1. Their main result, Theorem 4.4 (+ Cor 5.1), show that as n, p -> inf, the difference in equalized odds (test-time) is equivalent to the difference in $\widehat E[\hat a | s]$, where $\widehat \alpha$ is the representer coefficients of the learned RKHS estimator.
    - Note that the empirical fairness constraint requires the difference in $\widehat E[\hat a | y=y, s]$ be zero for all y. This is different from the above.
2. It is then shown in Theorem 5.2 that, in the limit (p > n, and p, n -> inf), if fairness is to be achieved via enforcing the empirical fairness constraint, it must be that the learned classifier performs no better than random guess.
3. To address this issue, they propose adding a term in the fairness constraint (with tunable parameters $\omega$; section 3.1); they show that there exists parameter settings such that the fair classifier is not ruled out. Then, they show that in the limit, achieving fairness via this "bias-overcompensated" constraint yields non-trivial classifiers.
    - The proposed fix is counter-intuitive at first (because it means not achieving empirical fairness), but the realization is that we are aiming for test-time fairness, and the fix derived based on the domain knowledge of this specific data/model/problem settings.
4. To estimate/tune the parameters, they propose searching over the grid and choosing the setting that minimizes the gap in the empirical $\widehat E[\hat a | s]$.

**Compliance With Llm Reviewing Policy:**

Affirmed.

**Final Justification:**

I retain my original rating because the limitations remain.

**Key Questions For Authors:**

see weaknesses

**Limitations:**

yes

**Strengths And Weaknesses:**

**Strengths.**

- Albeit requiring a noisy-feature and overparameterized setting, the paper reveals a subtle yet interesting failure mode of just applying the empirical equalized odds constraint in ERM.
- The theoretical results look sound, and are backed by empirical evidence.
- The study is thorough, including an ablation study in section 6 on "why not just minimize the empirical gap of $\widehat E[\hat a | s]$".

**Weaknesses.**

While there is technical merit and novelty (based on which the reviewer will score), its ultimate impact may be limited by the narrowness of the theoretical setting:

- The failure mode is only revealed in the noisy-feature + overparameterized setting, with linear (RKHS) models. And the subsequent analyses and remedy are only studied under the same setting.
- It is unclear whether the proposed "fix" is useful in general settings. It may be nice to empirical evaluate whether the proposed "fix" would hold under general (non-linear) models; under overparameterized settings.

---

> ### Author Rebuttal · Authors · 2026-03-30
>
> We sincerely thank the reviewer for the time invested in our paper. Please find below our response.
>
> W1: The failure mode is only revealed in the noisy-feature + overparameterized setting, with linear (RKHS) models. And the subsequent analyses and remedy are only studied under the same setting.
>
> We thank the reviewer for this important comment. We agree that the current theoretical analysis is conducted in a restricted setting: a spurious-feature data model, a proportional overparameterized regime, and an RKHS formulation that allows a tractable convex program in representer space. This scope is explicit in the paper. The studied spurious-feature data model was shown in previous work to exhibit bias amplification in the overparameterized regime, therefore appropriate for our investigation. The proportional regime allows an exact characterization of the learning results given by overparameterized models. Our analyis, like others in the proportional regime, requires a specification of the learning model, as it points to different learning mechanisms, therefore resulting in different characterizations. However we would like to point out that the failure of overparametrized fair ERM is not specific to our setting, as discussed in Section 1-1 and illustrated by related work of Section 2 (col 2 lines 110-123). Our theory advances the (primarily empirical) study of this issue by demonstrating its persistency in regularized solutions, tracing the problem to the unbalanced representer coefficients, and establishing bias overcompensation as effective remedy.
>
>
> W2: It is unclear whether the proposed "fix" is useful in general settings. It may be nice to empirical evaluate whether the proposed "fix" would hold under general (non-linear) models; under overparameterized settings.
>
> This work is not meant to claim a universal theorem for fairness intervention, but to contribute to a better understanding of bias mitigation via in-processing for large overparametrized models. In the provided Color-MNIST experiment, our method is tested using a two-layer neural network with a random feature mapping at the hidden layer. In that experiment, the same result is observed: classical fair ERM remains biased, while the proposed overcompensation procedure improves the group TPR/FPR gaps. It would indeed be interesting to extend to more general nonlinear overparameterized models, such as neural networks with feature learning at hidden layers. This would constitute a substantial deviation from our setup, and warrant an independent investigation in our view. A possible direction is adopting the framework of Dandi et al. (2024) to analyze fair ERM with one-step feature learning at the single hidden layer of neural networks. We will mention this approach in the revised version.
>
> Dandi, Y., Krzakala, F., Loureiro, B., Pesce, L., & Stephan, L. (2024). How two-layer neural networks learn, one (giant) step at a time. Journal of Machine Learning Research, 25(349), 1-65.

---

> > ### Author Rebuttal · Reviewer_AkbN · 2026-04-04
> >
> > I thank the author for the response; the limitations of the work still stands: that both theory and the experiments are limited to the linear setting. I would maintain my current score for the novelty in the theoretical contributions.

---

> > > ### Author Response · Authors · 2026-04-05
> > >
> > > Thank you for acknowledging the theoretical novelty of our paper. The extension to more general non-linear models is indeed important and will be discussed in the revised version. We would like to add that the previous analyses of Subramonian et al. (2024); Bombari & Mondelli (2025) , which were carried out in the same proportional regime as ours, focused also on linear models. These analyses demonstrated bias amplification for linear regression under spurious correlations, while ours studied bias mitigation by fair ERM in a classification setting.

---

### Official Review · Reviewer_hUm5 · 2026-03-12

**Soundness:** 3
**Presentation:** 2
**Significance:** 3
**Originality:** 3
**Overall Recommendation:** 5
**Confidence:** 2

**Summary:**

This paper is mostly a theoretical paper analysing why fair empirical risk minimisation fails in the over-parametrised regime and proposing a method to correct for this by modifying the fairness constraint. They also provide a search interval to identify how much to overcompensate the disadvantaged sensitive group, and show some empirical evidence for their findings in COLOR-MNIST experiments.

**Compliance With Llm Reviewing Policy:**

Affirmed.

**Final Justification:**

I recommend acceptance for this paper (5) as I think they tackle an original and important problem, and the paper is well presented, with a logical argument. The answers to my questions in the rebuttal were well thought out and mentioned interesting points, and the authors also said they would discuss links to similar empirical findings more. Again, my confidence is low as this is somewhat far from my expertise level.

**Key Questions For Authors:**

1. Empirical methods have tackled this problem by calculating e.g., loss on a held-out validation set on which the model hasn't overfit to. What do you think of this? How would this fit into your framework?
2. How would your theory extend to non-binary sensitive attributes?
3. Is this only valid for bias in the form of spurious correlations between an attribute and a label?
4. How do you explain that empirically, some bias mitigation methods do work even in the overparametrised regime?
5. How would this extend to non-linear losses?

**Limitations:**

Yes although I think they could expand on this, at the very least in the appendix.

**Strengths And Weaknesses:**

Strengths:
1. Bias mitigation in the overparametrised setting is an important problem, that, in my view, has not been given enough attention.
2. Although I did not review the theory in great detail, the authors seem to present a rigorous argument with well thought-out theory following an established line of work.

Weaknesses:
1. I think the authors should further motivate the empirical need for their work, explaining in more detail why bias mitigation in the overparametrised regime is an important issue. More empirical works could be discussed, for instance [1] and [2]
2. The overall clarity of the paper could be improved, I found some sections difficult to follow.
3. Math mode should be used consistently, eg col 2 lines 147 onwards.
4. I understand the core of the paper is theoretical, but given that overparametrised models are ubiquitous today, especially for larger models, it would be add to the paper to include further experiments in real data scenarios.

[1] FAIRTUNE: OPTIMIZING PARAMETER EFFICIENT FINE TUNING FOR FAIRNESS IN MEDICAL IMAGE ANALYSIS, ICLR 2024.
[2] Leveling Down in Computer Vision: Pareto Inefficiencies in Fair Deep Classifiers, CVPR 2022.

---

> ### Author Rebuttal · Authors · 2026-03-30
>
> We sincerely thank the reviewer for the time invested in our paper. Please find below our response.
>
> W1: We thank the reviewer for providing these two references that attest to the practical challenge of learning fair models in the overparametrised regime. We will add them in the revised version.
>
> W2: To help grasp the implications of our analysis, we have tried to discuss the theoretical results and explain the key steps for deriving the conclusions. We would be grateful for any suggestion to improve the clarity.
>
> W3: We deliberately use \textnormal{...} to distinguish random variables.
>
> W4: Indeed this paper is intended as a first contribution towards a more comprehensive theory of bias mitigation in the overparameterized regime. As our analysis already revealed several important messages, we chose to leave more involved practical settings to future investigation.
>
> Q1: When used for hyperparameter tuning, a held-out validation set can be viewed as a direct finite-sample estimator of the population fairness gap, and therefore as an empirical surrogate for the criterion that our theory approximates through balanced representer coefficients. Our approach differs in that it avoids using sample for hyperparameter tuning, which enables to use all the available data for the training.
>
> When used to calculate the fairness constraint integrated into the learning optimization, a held-out validation set might mitigate the overfitting problem to a certain extent. But in this case the optimization solution remains dependent of the held-out validation set. Therefore satisfying the fairness constraint on the held-out validation set would not suggest generalization of fairness to new data instances, specially when the number $n$ of available examples is only comparably large to the dimension $p$ of the learning model as in the overparametrized regime considered in our analysis.
>
> Q2: It is true that our analysis can be extended to the multi-group or continuous case. For the multi-group case, we could consider different patterns, each associated to one group, and the fairness constraint to be applied to all possible pairwise decomposition with bias overcompensation for the relatively underprivileged group. For a continuous sensitive attribute, as in the binary case equal average scores imply zero correlation between the target and the sensitive variable, we could interpret it as a zero correlation constraint for continuous sensitive attributes, then the bias overcompensation would be enforced by imposing negative correlation. We will add these remarks in the revised version.
>
> Q3: Yes this is not valid for any bias but for this version of spurious correlation bias which is a major phenomenon in ML. Actually,  an overparameterized model can often reach zero training error even if it uses the “wrong” signal. In datasets where a majority group exhibits a strong but non-causal correlation between a feature and the label, an overparameterized model can choose a solution that relies on that spurious feature for the majority and simply memorizes the minority exceptions. This improves average accuracy while hurting the groups for which the spurious association does not hold, violating fairness as highlighted by many papers such as Sagawal et al. (ICML 2020) for instance.
>  The key message is that interpolation makes the training fairness constraint too easy to satisfy while the test score retains a group-dependent shift; Hence fairness is restored only when that shift is neutralized.
>
> Q4: Our theory only rules out a specific class of in-processing techniques: classical fair ERM that cancels the fairness gap on the training set. An important discovery of our analysis is that the effectiveness of classical fair ERM can be restored through bias overcompensation. The method we propose in Algorithm 1 successfully produces fair results in the overparametrized regime. Our analysis does not study pre-processing and post-processing techniques, which can still work in the overparametrized regime. Comparing with these approaches would give rise to interesting future works. We will comment on this perspective in the revised version.
>
> Q5: The Assumption 4.3 requires the loss to be strictly convex and three-times differentiable and we consider both logistic and square losses as examples, the main theory is therefore not restricted to square loss. If the question refers to the fairness constraint, we have explicitly chosen $g$ to be linear, as it recovers several baselines of fair ERM and preserves the convexity of the constrained optimization. Moreover, as stated in Proposition 3.2, in the classical regime where $n\gg p$ it is with linear $g$ that fairness is achieved for the spurious-feature model considered in our analysis (following prior studies of Sagawa et al. (2020); Wald et al. (2022)). Changing to a non-linear $g$ might cause fair ERM to fail on this data model even in the classical regime where $n\gg p$.

---

> > ### Author Rebuttal · Reviewer_hUm5 · 2026-04-02
> >
> > Thanks for your comprehensive answers. I think the answers to Q1-Q4 are very interesting and would be worth including in the paper. After reading the answers to my questions and the other reviewers comments I have decided to raise my score to a 5. I however maintain that my confidence is low as I did not review the theory in detail.

---

> > > ### Author Response · Authors · 2026-04-05
> > >
> > > We are glad to know that you appreciate our reply and will increase the score. We will make sure to include there remarks in the revised version as suggested. Thank you again for the helpful comments and interesting questions.

---

### Decision · Program_Chairs · 2026-04-30

**Decision:**

Accept (regular)

**Comment:**

The reviewers agree the paper addresses the important problem of bias mitigation in an overparameterised setting with good theoretical results and empirical evidence. There are also concerns on the motivation, scope, presentation, experiments, and related work.

During the rebuttal, the authors provided responses that addressed most of the reviewer concerns. One reviewer pointed out that the limitation to linear models still remains, but the authors showed how previous analyses also use the same setup, which makes it reasonable. As the same reviewer mentioned, the technical merit and novelty are the strong points of the paper.

Since all the scores are clearly positive, the recommendation is to accept the paper.